# Harnessing strong metal–support interactions via a reverse route

Peiwen Wu [1,2,10], Shuai Tan[1,10], Jisue Moon [1], Zihao Yan[3], Victor Fung[4], Na Li[5,6], Shi-Ze Yang [1],
Yongqiang Cheng[7], Carter W. Abney[1], Zili Wu[1], Aditya Savara [1], Ayyoub M. Momen[8], De-en Jiang [4],
Dong Su [5], Huaming Li[2], Wenshuai Zhu [2✉], Sheng Dai [1,9✉] & Huiyuan Zhu [1,3✉]

Engineering strong metal–support interactions (SMSI) is an effective strategy for tuning structures and performances of supported metal catalysts but induces poor exposure of active sites. Here, we demonstrate a strong metal–support interaction via a reverse route (SMSIR) by starting from the final morphology of SMSI (fully-encapsulated core–shell structure) to obtain the intermediate state with desirable exposure of metal sites. Using core–shell nanoparticles (NPs) as a building block, the Pd–FeO$_x$ NPs are transformed into a porous yolk–shell structure along with the formation of SMSIR upon treatment under a reductive atmosphere. The final structure, denoted as Pd–Fe$_3$O$_4$–H, exhibits excellent catalytic performance in semi-hydrogenation of acetylene with 100% conversion and 85.1% selectivity to ethylene at 80 °C. Detailed electron microscopic and spectroscopic experiments coupled with computational modeling demonstrate that the compelling performance stems from the SMSIR, favoring the formation of surface hydrogen on Pd instead of hydride.

[1] Chemical Sciences Division, Oak Ridge National Laboratory, Oak Ridge, TN 37831, USA. [2] School of Chemistry & Chemical Engineering, Jiangsu University, Zhenjiang 212013, China. [3] Department of Chemical Engineering, Virginia Polytechnic Institute and State University, Blacksburg, VA 24061, USA. [4] Department of Chemistry, University of California, Riverside, CA 92521, USA. [5] Center for Functional Nanomaterials, Brookhaven National Laboratory, Upton, NY 11973, USA. [6] Frontier Institute of Science and Technology, Xi'an Jiaotong University, Xi'an 710054, China. [7] Neutron Scattering Division, Oak Ridge National Laboratory, Oak Ridge, TN 37831, USA. [8] Energy and Transportation Science Division, Oak Ridge National Laboratory, Oak Ridge, TN 37831, USA. [9] Joint Institute for Advanced Materials, University of Tennessee, Knoxville, TN 37996, USA. [10] These authors contributed equally: Peiwen Wu, Shuai Tan. ✉email: zhuws@ujs.edu.cn; dais@ornl.gov; huiyuanz@vt.edu

Supported metal catalysts have long been recognized as the most important group of heterogeneous catalysts for fundamental investigations and modern chemical industries[1–4]. Conventionally, these catalysts are synthesized by anchoring the active metal nanoparticles (NPs) onto certain high-surface-area supports to increase the dispersion of catalytically active sites and stabilize the metal against leaching[5–7]. Subsequently, the metal–support interface is constructed. Such an interface provides synergistic properties to regulate catalysis by modifying the electronic (charge transfer between the metal sites and the support) and/or geometric (decoration or coverage of metal sites by the support) parameters, and also by modulating the reaction pathways, e.g., lattice oxygen in oxide supports may directly participate in catalytic reactions[7]; multicomponent interfaces can enable tandem reaction pathways that do not exist on single-component active sites[8,9].

As a classic prototype in metal–support interactions, the strong metal–support interaction (SMSI) has been defined as the encapsulation of NPs, usually group VIII metals, by partially reduced oxide supports during high-temperature hydrogen ($H_2$) treatment[10,11]. Since the very first discovery of SMSI by Tauster et al.[12–14], SMSI has been widely exploited to tune catalytic performances of group VIII NPs by engineering geometric and/or electronic structures of these metal sites. For example, the adsorption of $H_2$ or CO on Pd was extremely suppressed upon the formation of SMSI (refs. [13,15]), suggesting that the active metal sites were largely covered by support, which altered the geometric ensembles and improved the thermal stability of Pd catalysts. Meanwhile, because the reducible oxide support, e.g., $TiO_2$, $Co_3O_4$, $CeO_2$, and $Nb_2O_5$, is partially reduced to the structure with a nonstoichiometric oxygen concentration during the reductive annealing, electron transfer between metal NPs and oxide supports was detected[16–19]. Under extreme conditions, the formation of intermetallic structure of the supported metal and metal cations in the supporting oxide was observed[20,21].

Despite these fascinating interfacial properties in SMSI, the formation of SMSI is restricted to specific combinations of elements, i.e., group VIII metals with high surface energy and transition metal oxides with low surface energy. Consequently, it is extremely challenging for some metals, e.g., Au, to manifest SMSI due to their low work function and surface energy[15,17,22].

Efforts have been devoted in hope of expanding upon the conventional SMSI. One critical element in this pursuit is switching the high-temperature treatment in $H_2$ into other conditions and thereby changes the mechanistic pathways for the formation of SMSI. For example, Wang et al. reported SMSI formation between Au NPs and $TiO_2$ induced by melamine under an oxidative atmosphere. With the formation of SMSI, the Au NPs were encapsulated by a permeable $TiO_x$ thin layer, making the Au NPs ultrastable at 800 °C (ref. [23]). Xiao et al. reported a wet chemistry approach to construct SMSI in aqueous solution at room temperature, which was realized by engineering redox interactions between metals and supports. This strategy was applicable to Au, Pt, Pd, and Rh (ref. [15]). Christopher et al. developed a strongly bounded-adsorbate-mediated strategy to construct SMSI between Rh and $TiO_2$ through high-temperature treatment in the mixture of $CO_2$ and $H_2$ (ref. [24]). Zhang et al. engineered the SMSI between Au NPs and hydroxyapatite by treating the Au NP–hydroxyapatite composite in the air at high temperatures[17]. Although progress has been made in expanding the boundaries of SMSI, one inevitable issue associated with the conventional SMSI is that upon high-temperature treatment the encapsulation process immediately and uncontrollably takes place, resulting in limited exposure of active sites[25]. In the ideal scenario, the oxide coverage on the metal surface needs to be thin and permeable to small molecules, while still fully encapsulating metal NPs to prevent the dissolution, disintegration, and aggregation of active sites during catalysis.

We recently reported that voids and cavity space can be developed in metal–metal oxide core–shell NPs in response to $H_2$ treatment at 200 °C (ref. [26]). This observation combined with the current issues in conventional SMSI motivated us to develop alternative routes to metal–support interactions. Here, we denote this type of structural rearrangement as the strong metal–support interaction via a reverse route (SMSIR). Specifically, we start from the final morphology of SMSI (full encapsulation) and end in the intermediate state with partial exposure of metal sites (Fig. 1). As a proof of concept, we demonstrate that the core–shell Pd–$FeO_x$ NPs can be restructured into a porous yolk–shell structure after optimized reductive annealing (Pd–$Fe_3O_4$–H). Characterizations reveal that Pd atoms gradually migrate into the $Fe_3O_4$ lattice and electron is partially transferred from Pd to $Fe_3O_4$. The

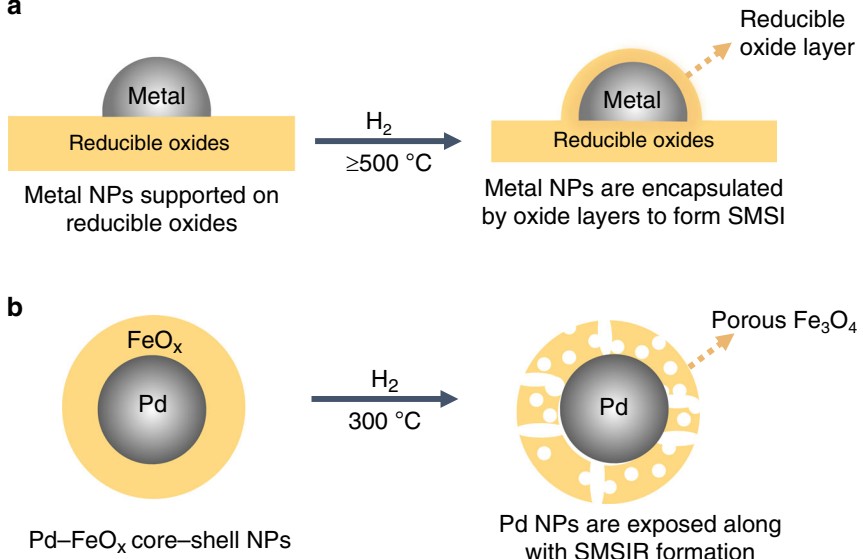

**Fig. 1 Schematic illustration of the formation of strong interactions. a** The conventional SMSI formation process. **b** The SMSIR formation process in this work.

Pd–Fe$_3$O$_4$–H shows 100% conversion and 85.1% selectivity in the acetylene (C$_2$H$_2$) semi-hydrogenation at atmospheric pressure and a mild reaction temperature of 80 °C. Further investigations demonstrate that the Pd–Fe$_3$O$_4$–H engineered by the SMSIR alleviates the strong H$_2$ adsorption on Pd sites, in favor of the formation of surface hydrogen (surface-H) instead of hydride during the hydrogenation of C$_2$H$_2$ to C$_2$H$_4$. Our results on the scenario of engineering SMSIR can help to circumvent the current limits in metal–support interfaces, expanding the boundaries of conventional SMSI, and providing opportunities to rationally maneuver structure-dependent catalytic outcomes.

## Results

**Synthesis and characterization.** Details of the material synthesis can be found in the "Methods" section. Briefly, the Pd NPs with a size of 5.5 ± 0.5 nm (Supplementary Fig. 1) were prepared by the reduction of palladium (II) acetylacetonate (Pd(acac)$_2$) in oleylamine (OAM) as modified from a previous report[27]. The core–shell Pd–FeO$_x$ NPs were obtained by a seed-mediated growth method with the pre-made Pd NPs as the seeds and iron (III) acetylacetonate, as the iron precursor that nucleated on Pd surface, forming an iron oxide shell. The pristine core–shell sample was denoted as Pd–FeO$_x$ NPs (Supplementary Figs. 2 and 3). The SMSIR was constructed by treating the Pd–FeO$_x$ NPs at 300 °C in a gas mixture of H$_2$ and argon (Ar; 4 vol.% of H$_2$), and the sample was named as Pd–Fe$_3$O$_4$–H. As a comparison, the Pd–FeO$_x$ NPs were treated in the air at 300 °C to obtain the structure without SMSIR (Pd–Fe$_3$O$_4$–A).

X-ray diffraction (XRD) was performed to determine the crystal structures of the samples. As shown in the XRD patterns of pristine Pd–FeO$_x$ and Pd–Fe$_3$O$_4$–A (Supplementary Fig. 4), characteristic peaks at $2\theta = 40.1°$ with very low intensities were detected, which can be assigned to the (111) peak of face-centered cubic (fcc) Pd. No additional peaks in the XRD patterns can be found, indicating the amorphous nature of iron oxide shell in both pristine Pd–FeO$_x$ and Pd–Fe$_3$O$_4$–A. On the contrary, in the XRD pattern of Pd–Fe$_3$O$_4$–H, the intensity of Pd (111) peak increases remarkably, and a series of characteristic peaks at $2\theta = 30.6°$, 35.9°, 43.5°, 53.9°, 57.3°, 63.0°, and 74.3° are clearly observed, which can be assigned to (220), (311), (400), (422), (511), (440), and (533) lattices of $\gamma$-Fe$_3$O$_4$. The XRD characterization indicates that annealing in the reductive atmosphere may facilitate the spatial redistribution of grains in the oxide shell, and promotes the crystallization of Pd and iron oxides, consistent with our previous report[26].

The aberration-corrected high-angle annular dark-field scanning transmission electron microscopy (HAADF-STEM) images of the Pd–Fe$_3$O$_4$–H in Fig. 2a show that the core–shell structure of pristine Pd–FeO$_x$ NPs evolved into a unique porous yolk–shell structure after reductive annealing at 300 °C. Magnified HR-STEM images of Pd–Fe$_3$O$_4$–H in Fig. 2b, c demonstrate a lattice parameter of 0.217 nm in the core, corresponding to the (111) plane of Pd, and lattice parameters of 0.251 and 0.146 nm in the shell, corresponding to the (311) and (440) planes of Fe$_3$O$_4$. More interestingly, the magnified HR-STEM image in Fig. 2d shows that there are abundant voids, i.e., lattice vacancies, in the Fe$_3$O$_4$ shells (marked in yellow circles). To analyze the pore distribution, the pore sizes were determined by averaging pore sizes in multiple HR-STEM images (Supplementary Fig. 5). The majority of these pores on the Fe$_3$O$_4$ shell are micropores with an average pore size of 0.73 nm. Furthermore, electron energy loss spectroscopy (EELS) mapping of Pd–Fe$_3$O$_4$–H in Fig. 2e–i depict a yolk–shell-like structure of Pd yolk and Fe$_3$O$_4$ shell with numerous voids. In contrast, for Pd–Fe$_3$O$_4$–A, no significant voids were detected in the Fe$_3$O$_4$ shells and the intact core–shell structure was retained (Supplementary Fig. 6). It is known that the reducible metal oxides can be partially reduced after high-temperature treatment in H$_2$ (ref. [28]). H$_2$ reacts with these oxides to produce water and generate oxygen vacancies in the oxide matrix. This process can be further facilitated by platinum-group metal NPs supported on those oxides through a H$_2$ spillover process[29,30]. In the meantime, the crystallization of oxide shell could promote the rearrangement of

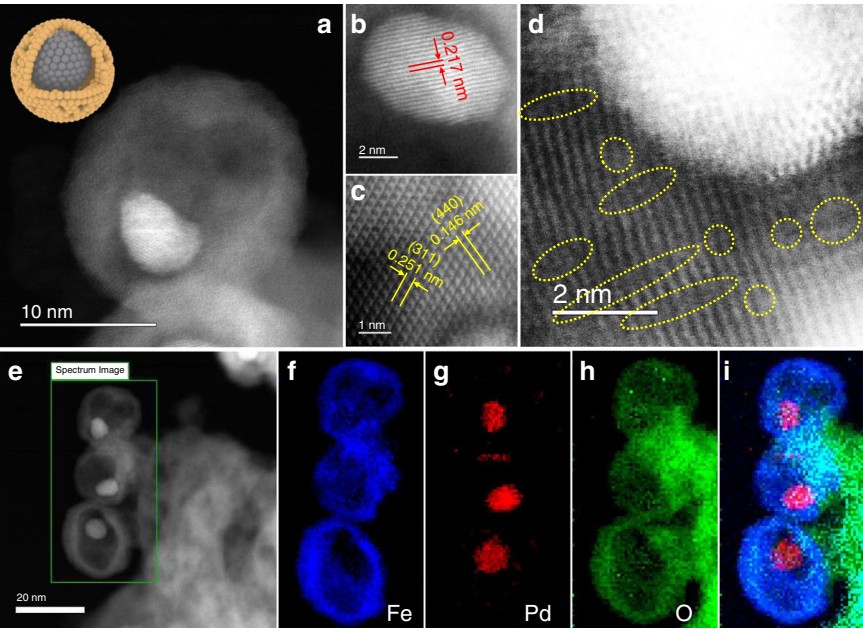

**Fig. 2 STEM characterization of the Pd–Fe$_3$O$_4$–H sample. a** HAADF-STEM image of Pd–Fe$_3$O$_4$–H and schematic illustration of the structure (the yellow shell stands for Fe$_3$O$_4$, and grey core stands for Pd); **b** HR-STEM images of Pd core, **c** Fe$_3$O$_4$ shell, and **d** Pd–Fe$_3$O$_4$–H (voids in oxide shell are marked in yellow circles); **e** STEM image of Pd–Fe$_3$O$_4$–H at lower magnification; **f–i** corresponding EELS elemental mappings of the selected section in **e**; **f** Fe, **g** Pd, **h** O, and **i** overlapped figure.

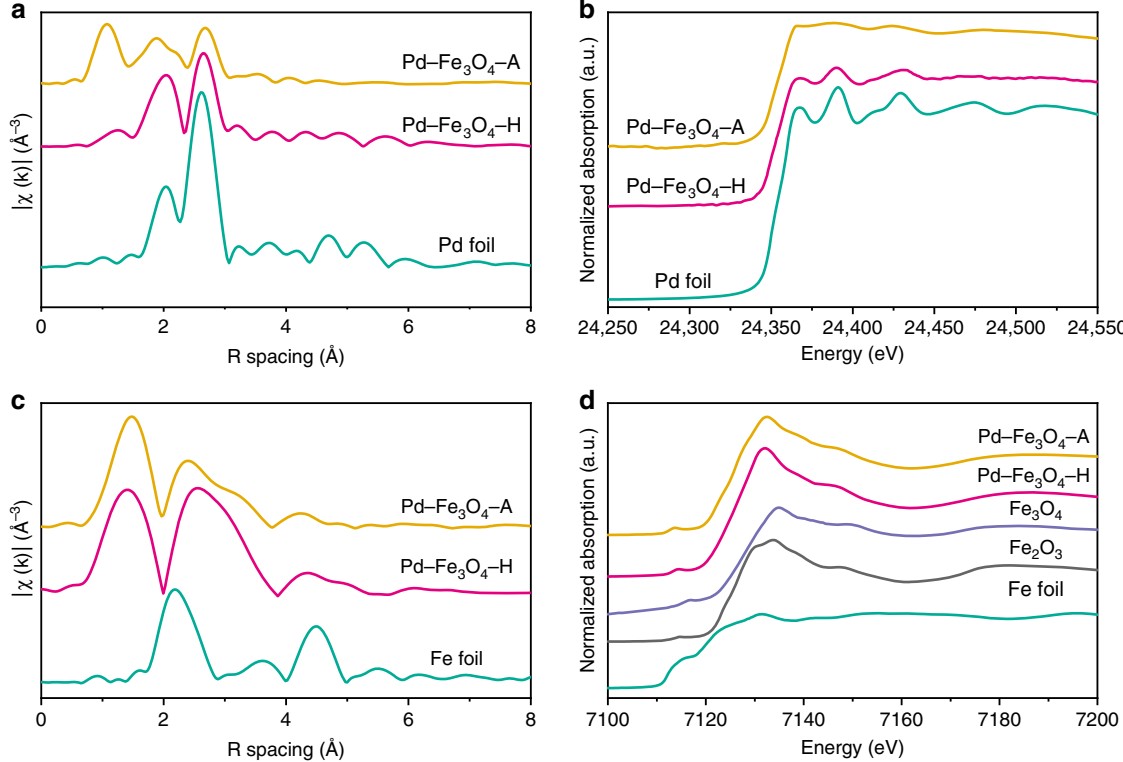

**Fig. 3 XAFS characterization of prepared samples and reference samples. a** Pd K-edge EXAFS; **b** Pd K-edge XANES; **c** Fe K-edge EXAFS; and **d** Fe K-edge XANES. Source data are provided as a Source data file.

atoms, alter the distribution of oxide grains to expand the generated oxygen vacancies, and finally develop voids and cavity space in the structure.

**XAFS characterization and simulations**. To understand the coordination environments of Pd–Fe$_3$O$_4$ structures, X-ray absorption near-edge spectroscopy (XANES) and extended X-ray absorption fine structure (EXAFS) were performed (Fig. 3, Supplementary Tables 1 and 2). The chemical states of Pd in Pd–Fe$_3$O$_4$–H and Pd–Fe$_3$O$_4$–A samples were investigated in Pd K-edge EXAFS and XANES, and Pd foil was employed as a reference (Fig. 3a, b, Supplementary Table 1). The Pd in Pd–Fe$_3$O$_4$–H mainly exists in the form of metallic Pd$^0$, while in Pd–Fe$_3$O$_4$–A, Pd demonstrates an oxidized feature to some extent. To determine the chemical states and structures of iron oxide shells, the Fe K-edge EXAFS (Fig. 3c), corresponding fitting (Supplementary Table 2), the Fe K-edge XANES (Fig. 3d), and the Fe K-edge first derivative XANES (Supplementary Fig. 7) were collected and analyzed. Compared with the Fe$_3$O$_4$ and Fe$_2$O$_3$ references, the Fe K-edge XANES and the Fe K-edge first derivative XANES indicated that the oxide shell in Pd–Fe$_3$O$_4$–H was similar to Fe$_3$O$_4$, while the oxide shell in Pd–Fe$_3$O$_4$–A possessed a partially oxidized Fe$_3$O$_4$ feature (Fig. 3d, Supplementary Fig. 7).

Because Pd–Pd and Pd–Fe bond lengths are similar, it is hard to visualize the difference between these two bonds with Fourier transform results. In this regard, the wavelet transform (WT) EXAFS as a powerful technique was employed to distinguish these two bonds in our samples. It can be clearly seen from Fig. 4a, b that compared with the standard WT EXAFS images for Pd–Fe, Pd–O, Pd–Pd, and Pd foil (Supplementary Fig. 8), the Pd in Pd–Fe$_3$O–H remains to be the metallic Pd$^0$ state and Fe–Pd bond emerges (Fig. 4a). In contrast, for the Pd–Fe$_3$O$_4$–A sample (Fig. 4b), the result demonstrates an oxidized feature with the formation of the Pd–O bond, indicating that the Pd may be

slightly oxidized by air, consistent with our EXAFS and XANES results in Fig. 3.

Density functional theory (DFT) simulations combined with the EXAFS curve fitting were carried out to provide more insight into the crystal structure of iron oxide, and the interactions between Pd and Fe$_3$O$_4$. First, a series of models including a Pd cluster atop the surfaces of Fe$_2$O$_3$ and Fe$_3$O$_4$, and a Pd cluster in oxygen vacancy of Fe$_2$O$_3$, and Fe$_3$O$_4$ surfaces were constructed and optimized by DFT in Supplementary Fig. 9, and the corresponding FEFF calculated scattering paths were also presented (Supplementary Tables 3–5). Then, the EXAFS curve fitting on the DFT-optimized structures (Supplementary Figs. 10 and 11, Supplementary Tables 6 and 7) of both Pd K-edge EXAFS and Fe K-edge EXAFS were obtained. It can be concluded from the results that the best-fitted structure of Pd–Fe$_3$O$_4$–H is that the Pd atoms intercalate into the Fe$_3$O$_4$ matrix (Fig. 4c, detailed optimizing process see Supplementary Fig. 10), indicating that the Pd enters into the Fe$_3$O$_4$ lattice, substituting an oxygen vacancy and tends to form the Fe–Pd bond with Fe in Fe$_3$O$_4$. This observation suggests that there exist strong interactions between Pd and Fe$_3$O$_4$ in Pd–Fe$_3$O$_4$–H. In contrast, the Pd–Fe$_3$O$_4$–A demonstrated a good match to the local geometry of Pd atoms situated on the surface of Fe$_3$O$_4$ (Fig. 4d, detailed optimizing process see Supplementary Fig. 11). This result may shed some light on the formation mechanism of this unique porous yolk–shell structure. Evidently, the reaction between H$_2$ molecules and O atoms in the Fe$_3$O$_4$ could generate oxygen vacancies in the structure upon evaporating the produced water. Meanwhile, the crystallization of the oxide shell and the new Fe–Pd bond formation could promote the rearrangement of oxide lattice and the mobility of Pd atoms, expanding the atom vacancies and developing cavity space in the structure.

**XPS and DRIFTS investigations**. To investigate the charge transfer with the formation of SMSIR, X-ray photoelectron

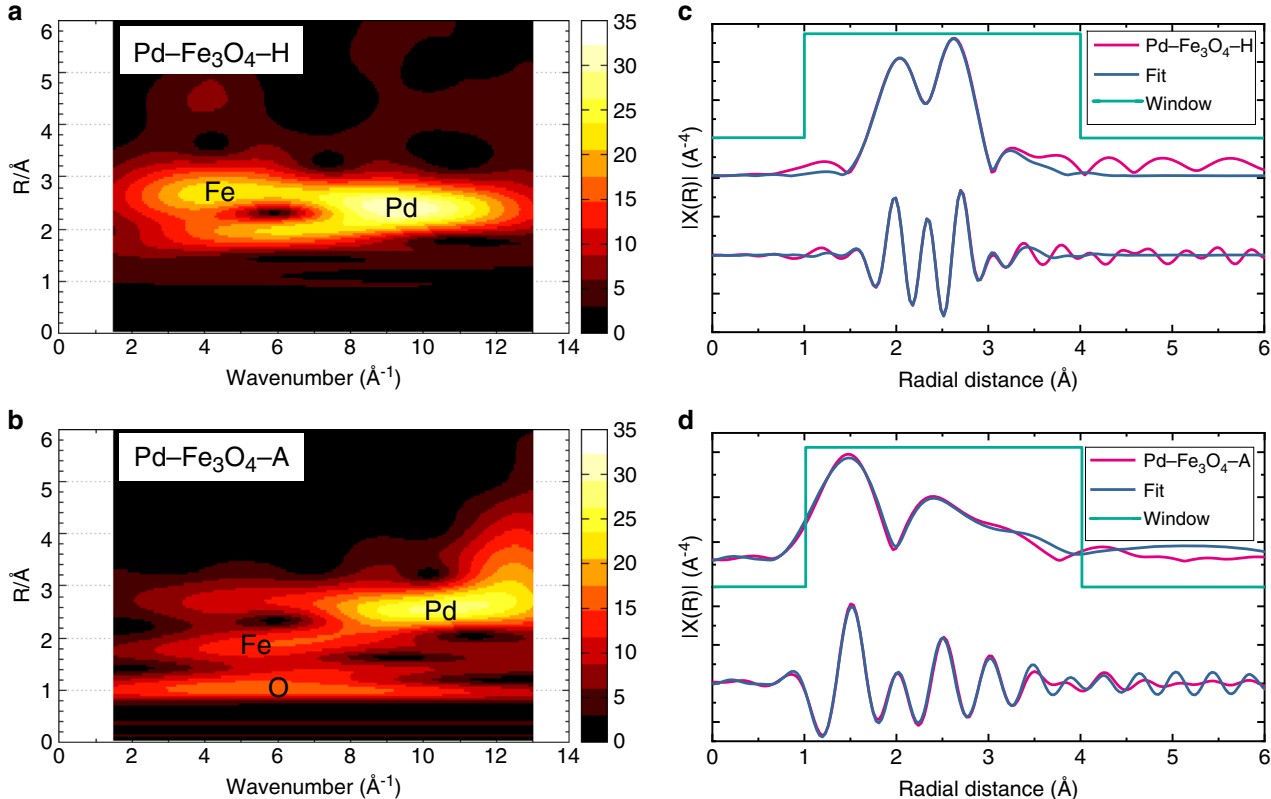

**Fig. 4 Pd K-edge WT EXAFS and EXAFS fitting using DFT-optimized results. a** Pd K-edge WT EXAFS of Pd-Fe$_3$O$_4$-H; **b** WT EXAFS of Pd-Fe$_3$O$_4$-A; **c** EXAFS curve fitting on DFT-optimized Pd-Fe$_3$O$_4$-H model; and **d** EXAFS curve fitting on DFT-optimized Pd-Fe$_3$O$_4$-A model. Source data are provided as a Source data file.

spectroscopy (XPS) and CO diffuse reflectance infrared Fourier transform spectroscopy (DRIFTS) were carried out and shown in Supplementary Figs. 12–14. In the high-resolution Pd 3$d$ XPS of Pd-FeO$_x$ NPs, only a Pd 3$d_{5/2}$ peak at 335.4 eV, being assigned to metallic Pd, was found[31,32]. When the Pd-FeO$_x$ NPs were treated in air at 300 °C, an additional Pd 3$d_{5/2}$ peak at 336.8 eV assigned to PdO, emerged in Supplementary Fig. 12. This observation is consistent with our XAFS results that the Pd in Pd-Fe$_3$O$_4$-A possesses an oxidized feature to some extent. Furthermore, a Pd 3$d_{5/2}$ shoulder peak at 336.2 eV in the high-resolution Pd 3$d$ XPS of Pd-Fe$_3$O$_4$-H was detected and assigned to the positively charged Pd (Pd$^{\delta+}$)[33], which is originated from intercalation of Pd into Fe$_3$O$_4$ matrix, leading to the strong interactions between Pd and Fe to form Pd-Fe bond[34], in accordance with the XAFS results. Meanwhile, in the high-resolution Fe 2$p$ XPS of Pd-Fe$_3$O$_4$-H (Supplementary Fig. 13), the peaks of Fe 2$p_{1/2}$ and Fe 2$p_{3/2}$ downshifted comparing with that of Pd-Fe$_3$O$_4$-A, further confirming that the charge transfers from Pd to Fe$_3$O$_4$ in Pd-Fe$_3$O$_4$-H. The XPS results indicate the formation of strong interactions and partial electron transfer between Pd and Fe$_3$O$_4$. The CO DRIFTS was further carried out. During the test, we found that the CO adsorption peak was weak. We, therefore, subtracted the pure gas-phase signal from each data set. As shown in Supplementary Fig. 14, a peak at ~2153 cm$^{-1}$ was detected both in the CO DRIFTS of Pd-Fe$_3$O$_4$-H and Pd-Fe$_3$O$_4$-A, which is assigned to Fe$^{3+}$-CO (ref. [35]). Due to the core–shell morphology of Pd-Fe$_3$O$_4$-A where Pd is fully encapsulated by Fe$_3$O$_4$, no obvious peak was detected in the CO DRIFTS of Pd-Fe$_3$O$_4$-A. In contrast, a very weak peak at 2102 cm$^{-1}$ can be seen in the CO DRIFTS of Pd-Fe$_3$O$_4$-H, which is assigned to the linear CO adsorption on metallic Pd (ref. [36]). More interestingly, an additional peak at 2134 cm$^{-1}$ can be found. Compared with the linear

CO adsorption on metallic Pd, this blueshifted peak is assigned to linear CO adsorption on positively charged Pd (CO-Pd$^{\delta+}$)[37]. Combined with the XPS analysis, this peak may be attributed to the linear CO adsorption on Pd$^{\delta+}$ in the newly emerged Pd-Fe bond of Pd-Fe$_3$O$_4$-H.

The Pd-Fe$_3$O$_4$-H sample was further re-treated in air at 300 °C to obtain the Pd-Fe$_3$O$_4$-Re sample, and characterized by TEM, CO DRIFTS, and XPS to determine the reversibility of SMSIR. As shown in the TEM image of the Pd-Fe$_3$O$_4$-Re (Supplementary Fig. 15), the sample still possesses a yolk–shell structure, but the voids are smaller than those of Pd-Fe$_3$O$_4$-H. The XPS of Pd-Fe$_3$O$_4$-Re (Supplementary Fig. 16) demonstrates three Pd states of metallic Pd, Pd$^{\delta+}$ in Pd-Fe bond, and PdO. The intensity of Pd$^{\delta+}$ peak is lower than that of Pd-Fe$_3$O$_4$-H (Supplementary Fig. 12), indicating the decrease of Pd$^{\delta+}$ concentration. The CO DRIFTS of Pd-Fe$_3$O$_4$-Re in Supplementary Fig. 17 shows that the intensity of CO-Pd$^{\delta+}$ became weaker than that in the CO DRIFTS of Pd-Fe$_3$O$_4$-H in Supplementary Fig. 14, suggesting the CO DRIFTS spectral feature is an intermediate state between Pd-Fe$_3$O$_4$-H and Pd-Fe$_3$O$_4$-A. The analysis of TEM, CO DRIFTS, and XPS together suggests that the SMSIR in this work is partially reversible.

**Catalytic performance.** The semi-hydrogenation of C$_2$H$_2$ to C$_2$H$_4$ is an important reaction in industrial purification of the C$_2$H$_4$ stream produced from naphtha cracking. Pd-based catalysts are mostly used for this reaction with a consensus that the selectivity is sensitive to the structure of the catalyst[38]. H$_2$ molecules that are weakly adsorbed onto the Pd surface to form surface-H and C$_2$H$_2$ molecules that are strongly adsorbed can lead to the production of C$_2$H$_4$, while the formation of hydride

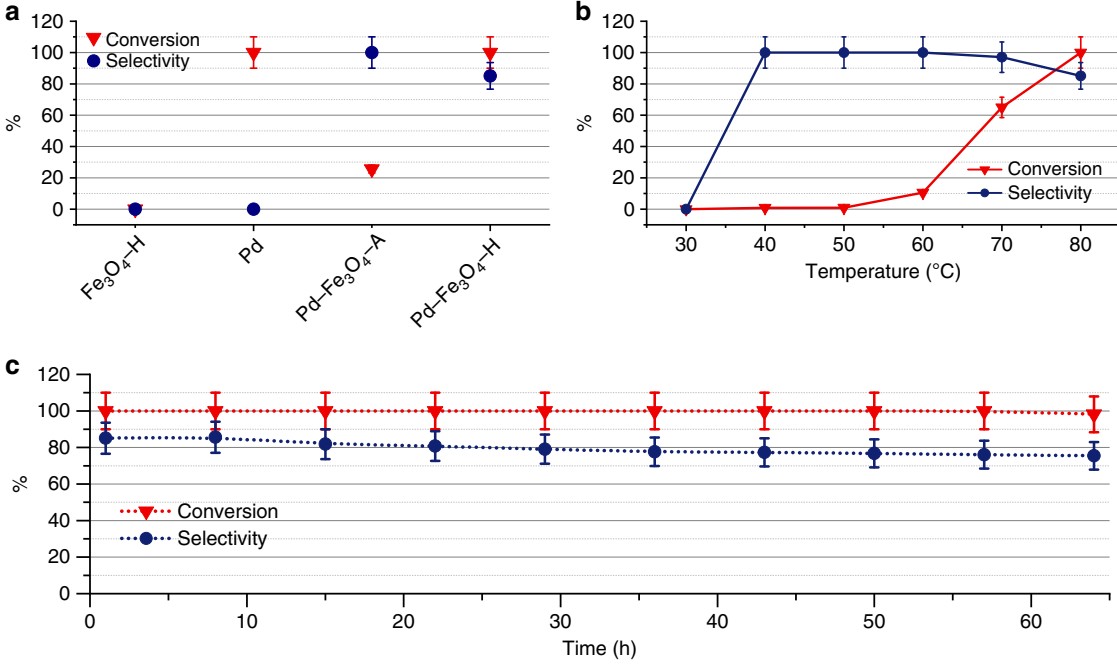

**Fig. 5 Catalytic performances of the prepared samples for the semi-hydrogenation of C₂H₂. a** The comparison of catalytic performance of different systems; **b** light-off curves of the Pd–Fe₃O₄–H catalyst; **c** stability of Pd–Fe₃O₄–H under high conversion. Reaction conditions: $m$ (catal.) = 15 mg; $v$ (gas) = 50 sccm (0.6 sccm C₂H₂, 3 sccm H₂, 46.4 sccm He). The reaction temperature in **a** and **c** is 80 °C. Error bars represent the instrumental error (±10 %). Source data are provided as a Source data file.

usually results in the total hydrogenation to ethane ($C_2H_6$). The adsorption of $H_2$ and $C_2H_2$ strongly relies on the structure of Pd catalysts. Herein, the hydrogenation of $C_2H_2$ was systematically investigated over the prepared catalysts to correlate their structural properties with the catalytic outcomes.

As shown in Fig. 5a, Pd NPs totally converted $C_2H_2$ to $C_2H_6$ without any selectivity toward $C_2H_4$ at 80 °C, while $Fe_3O_4$ NPs treated at 300 °C in a gas mixture of $H_2$ and Ar (4 vol.% of $H_2$; $Fe_3O_4$–H) barely demonstrated any catalytic activity for the hydrogenation of $C_2H_2$. The selectivity of $C_2H_2$ to $C_2H_4$ over the Pd–Fe₃O₄–A was 100%, but the conversion was only 25.6%. This observation can be mainly attributed to the core–shell structure of the Pd–Fe₃O₄–A that only exposes limited active Pd sites through the amorphous oxide shell, restricting the adsorption of the reactants. When the Pd–Fe₃O₄–H was employed in the semi-hydrogenation of $C_2H_2$ at 80 °C, the conversion was 100% and the selectivity was as high as 85.1%. The light-off curves of Pd–Fe₃O₄–H (Fig. 5b) demonstrate that the conversion of $C_2H_2$ increases with the increment of reaction temperature, while the selectivity toward $C_2H_4$ shows the opposite trend. To comprehensively compare the catalytic performance between the Pd–Fe₃O₄–H catalyst and previously reported values, the turnover frequency (TOF) was calculated based on the dispersion of Pd (obtained from $H_2$-pulse chemisorption in Supplementary Table 8). The TOF of Pd–Fe₃O₄–H was 6.46 s⁻¹, ~100-fold higher than those of a series of state-of-the-art single-atom catalysts at 80 °C (Supplementary Fig. 18), indicating that the Pd–Fe₃O₄–H demonstrated compelling catalytic performance for semi-hydrogenation of $C_2H_2$. Stability tests of the Pd–Fe₃O₄–H catalyst were further carried out under both the high and low conversion rates (Fig. 5c, Supplementary Fig. 19). Both results show that the Pd–Fe₃O₄–H catalyst was remarkably stable in the semi-hydrogenation of $C_2H_2$ to $C_2H_4$, which could be originated from the SMSIR between Pd and $Fe_3O_4$.

The formation of hydride in Pd-based catalysts is temperature-sensitive, and it dominates the total hydrogenation of $C_2H_2$

(refs. [38,39]). Hence, the dispersion of Pd is determined by $H_2$-pulse chemisorption (Supplementary Table 8) at various temperatures to examine the formation of hydride. For the reference Pd NPs (commercial 5 wt.% Pd/Al₂O₃), the dispersion was determined to be 7.6% at −130 °C (cold bath of isopentane and liquid $N_2$). The corresponding particle size was calculated to be 14.8 nm. However, $H_2$ uptake on Pd NPs increased significantly at 35 °C, and the estimated particle size decreased to 1.6 nm. This discrepancy can be attributed to the substantial formation of hydride on Pd NPs at higher temperature that interferes with the estimation of particle size[40]. In contrast, the Pd–Fe₃O₄–H sample demonstrated a dispersion of 26.7 and 24.4% at −130 and 35 °C, respectively. The corresponding particle sizes were calculated to be 4.2 and 4.6 nm, in agreement with the Pd core size from STEM investigations (Fig. 2). These observations indicate that the formation of hydride may be effectively inhibited in our Pd–Fe₃O₄–H catalyst with SMSIR, leading to a superior selectivity toward semi-hydrogenated products in the catalytic investigations.

**Control experiments.** A series of control samples obtained at different treating temperatures (T200, T300, and T400) were prepared (see "Methods" section) to determine the optimized condition for the formation of SMSIR. Here, the T300 stands for the Pd–Fe₃O₄–H sample with SMSIR. The structures and compositions of all catalysts are characterized by TEM, STEM (Supplementary Figs. 20 and 21), EXAFS, XANES, EXAFS curve fitting on DFT-optimized model (Supplementary Figs. 22–29, Supplementary Tables 9–11), and XRD patterns (Supplementary Fig. 30). The corresponding structures are summarized here: in the pristine core–shell Pd–FeOₓ sample, the core and shell were metallic Pd⁰ and amorphous $Fe_3O_4$. When the sample was treated in the air at high temperature (Pd–Fe₃O₄–A), the core–shell structures maintained with no obvious formation of voids. When the Pd–FeOₓ NPs sample was treated in $H_2$ at different temperatures, the core and shell crystallized into Pd⁰ and $Fe_3O_4$, respectively. As a result,

the T200 demonstrated a core–shell structure with fewer voids, T300 (Pd–Fe$_3$O$_4$–H) embraced a yolk–shell-like structure with numerous voids, and the T400 showed a heterostructure of Fe$_3$O$_4$ islands on Pd NPs. Especially, it can be seen from the EXAFS and corresponding fitting results of T300, i.e., Pd–Fe$_3$O$_4$–H, (Fig. 3c, Supplementary Table 2) that compared with the sample obtained at lower annealing temperatures (T200 sample; Supplementary Fig. 24, Supplementary Table 10), the Fe–O coordination number decreased but Fe–Fe coordination number remained stable in Pd–Fe$_3$O$_4$–H. This result further suggests that in Pd–Fe$_3$O$_4$–H, Pd may substitute the oxygen in iron oxide and form a new Fe–Pd bond, indicating the formation of strong interactions between Pd NPs and Fe$_3$O$_4$ shell. In addition, Pd–FeO$_x$ NPs with different shell thicknesses (STs) were also prepared. With the increment of STs, the samples were denoted as ST1 NPs, ST2 NPs (i.e., Pd–FeO$_x$ NPs), and ST3 NPs, and the NPs were further loaded onto $\gamma$-Al$_2$O$_3$ to obtain ST1, ST2 (i.e. Pd–Fe$_3$O$_4$–H), and ST3 (for characterizations see Supplementary Figs. 31–34). To help understand the structures of all prepared samples, a schematic diagram was presented in Supplementary Figs. 35 and 36.

To highlight the role of SMSIR in tuning the conversion and selectivity of C$_2$H$_2$ semi-hydrogenation, both sets of control samples were employed in the C$_2$H$_2$ semi-hydrogenation reaction. As shown in Supplementary Figs. 37 and 38, the Pd–Fe$_3$O$_4$–H, i.e., T300 and ST2, demonstrates the best catalytic performance. This result further reveals that the optimized ST and treating condition are essential to the formation of SMSIR for the promoted semi-hydrogenation of C$_2$H$_2$ to C$_2$H$_4$. Based on the structures of the catalysts, the different catalytic outcomes can be attributed to the following factors: (1) regarding the effect of annealing temperatures, all samples possess a similar Pd size, indicating that the difference of catalytic performance is not originated from the difference of particle sizes (Supplementary Fig. 39). In the T200-sample, there are fewer voids in the oxide shell and the T200 remains to be a core–shell structure, resulting in poor exposure of Pd active sites with limited activity. In the case of T400 sample, the core–shell structure is completely destroyed, and therefore the formation of hydrides turns to be favorable because of the loss of core–shell structural confinement; (2) for the effect of ST, a thicker shell in ST3 makes the Pd active sites less exposed. However, when the shell becomes too thin as the case in ST1, the structure cannot maintain a fully-encapsulated state, but rather more like a heterostructure with some iron oxide islands on Pd NPs. Consequently, Pd domains tend to form hydrides due to the lack of core–shell structural confinement effect.

**Reaction mechanism**. The reaction kinetics were further explored to understand the underlying mechanisms. As shown in Supplementary Fig. 40, the reaction order over C$_2$H$_2$ is calculated to be −1 (up to 2.5% atm partial pressure), roughly in agreement with Monnier's work with ~−0.7 reaction order[41], indicating the strong adsorption of C$_2$H$_2$ on the surface of Pd in the Pd–Fe$_3$O$_4$–H catalyst. There exists a debate regarding the H$_2$ reaction order. In general, the reaction order varies from ~0.5 (refs. [42,43]), ~1 (refs. [44,45]), and up to ~1.6 (ref. [46]). In our work, we found the reaction order of H$_2$ to be ~2 (up to 10% atm partial pressure). Such a positive dependence on H$_2$ partial pressure indicates a much weaker H$_2$ adsorption than previous studies. The temperature dependence of the Pd–Fe$_3$O$_4$–H sample was investigated at 1.2%/6% atm partial pressure of C$_2$H$_2$/H$_2$ (Supplementary Fig. 41). The apparent activation energy was found to be ~52.7 kJ mol$^{-1}$, in good agreement with Monnier's 12.1 kcal mol$^{-1}$ (ref. [41]) and Zhang's 52 kJ mol$^{-1}$ (ref. [45]).

The inelastic neutron scattering (INS) spectra of H$_2$ adsorption on Pd–Fe$_3$O$_4$–H and bulk Pd were presented in Fig. 6. The signal

of H$_2$-sorption behavior in Pd–Fe$_3$O$_4$–H is totally different from that in bulk PdH$_x$. In bulk PdH$_x$ sample, an evident signal of hydride was detected, while in Pd–Fe$_3$O$_4$–H, the signal of hydride was very weak[47,48]. The profile of the peak at 500 cm$^{-1}$ reflects the status of the hydride. Specifically, the sharp peak followed by a shoulder as seen in bulk PdH$_x$ is due to certain dispersion relation of optical phonons in 3D space, which results in this particular distribution of phonon states. When hydride is only formed at or near the surface, the 3D network is lacking, leading to the broad bump in the spectrum of our Pd–Fe$_3$O$_4$–H sample. The result indicates that only surface-H formed during the reaction process, consistent with our H$_2$-chemisorption results.

## Discussion

In this work, we reported a strategy to engineer the SMSI between Pd and Fe$_3$O$_4$ by using core–shell NPs as a building block through a reverse process of the formation of conventional SMSI, denoted as SMSIR. With the formation of SMSIR, the core–shell Pd–FeO$_x$ NPs was restructured into a unique porous yolk–shell structured Pd–Fe$_3$O$_4$–H, in favor of the exposure of Pd active sites. The Pd–Fe$_3$O$_4$–H with SMSIR demonstrated excellent catalytic performance in semi-hydrogenation of C$_2$H$_2$ to C$_2$H$_4$ with 100% conversion, 85.1% selectivity, and a high TOF of 6.46 s$^{-1}$ at the reaction temperature as low as 80 °C. XAFS investigations along with DFT simulations verified that the Pd atoms intercalate into the Fe$_3$O$_4$ matrix and form strong interactions. The electron transfer was probed by CO DRIFTS and XPS, suggesting that with the formation of SMSIR, electrons partially transfers from Pd to Fe$_3$O$_4$ shell. The optimized ST of Pd–FeO$_x$ NPs and annealing temperature were found to be essential to the formation of SMSIR. Detailed mechanistic investigations indicated that the SMSIR in Pd–Fe$_3$O$_4$–H alleviates the strong chemisorption of H$_2$ on Pd sites, prevents the formation of hydride, and consequently leads to a superior selectivity toward C$_2$H$_4$. This work not only develops a high-performance catalyst for semi-hydrogenation of C$_2$H$_2$ but also provides an approach for the construction of effective catalytic structures based on unconventional SMSI.

## Methods

**Chemicals**. Pd(acac)$_2$ (>99.99%), OAM (90%), tri-n-octylphosphine (TOP, AR.), hexane (AR.), ethanol (AR.), ferric (III) acetylacetonate (Fe(acac)$_3$, >99.99%), and $\gamma$-Al$_2$O$_3$ were obtained from Sigma Aldrich Corporate (USA) without further purification.

**Preparation of 4 nm Pd NPs**. The Pd NPs were prepared by a modified method from previous work as following[27]: 70 mg of Pd(acac)$_2$ was mixed with 15 mL of OAM in a 100 mL of four-neck flask under stirring. The mixture was then heated to 80 °C at a ramping rate of 5 °C min$^{-1}$ and kept for 1 h under the protection of N$_2$. A total of 0.5 mL of TOP was added to the solution. The mixture was further heated to 250 °C at a ramping rate of 5.6 °C min$^{-1}$, and kept at this temperature for another 1 h before cooling down to room temperature. Subsequently, the mixture was transferred to a 50 mL of centrifuge tube, and 30 mL of ethanol was added. The Pd NPs were separated by centrifugation at 4656 × g for 10 min. Then, the Pd NPs were redispersed in 10 mL of hexane, and precipitated and washed by adding 30 mL of ethanol for two times. Finally, the Pd NPs were dispersed in 10 mL of hexane for further use.

**Preparation of Pd–FeO$_x$ NPs**. A total of 110 mg of Fe(acac)$_3$ and 20 mL of OAM were added to a 100 mL four-neck flask. The mixture was heated to 90 °C at a ramping rate of 5 °C min$^{-1}$ in N$_2$. Subsequently, 12.5 mg of Pd NPs were added, followed by heating to 250 °C and kept there for 30 min. Afterward, the reaction temperature was raised to 300 °C and kept there for another 30 min before naturally cooling to room temperature. Then, 30 mL of ethanol was added to precipitate the Pd–FeO$_x$ NPs, and then centrifuged at 4656 × g for 10 min. The Pd–FeO$_x$ NPs was redispersed in 10 mL of hexane and washed by 30 mL of ethanol for two times. Finally, the Pd–FeO$_x$ NPs were dispersed in 10 mL of hexane.

**Preparation of FeO$_x$ NPs**. The synthesis of FeO$_x$ NPs is the same as the preparation of Pd–FeO$_x$ NPs, without adding Pd NPs.

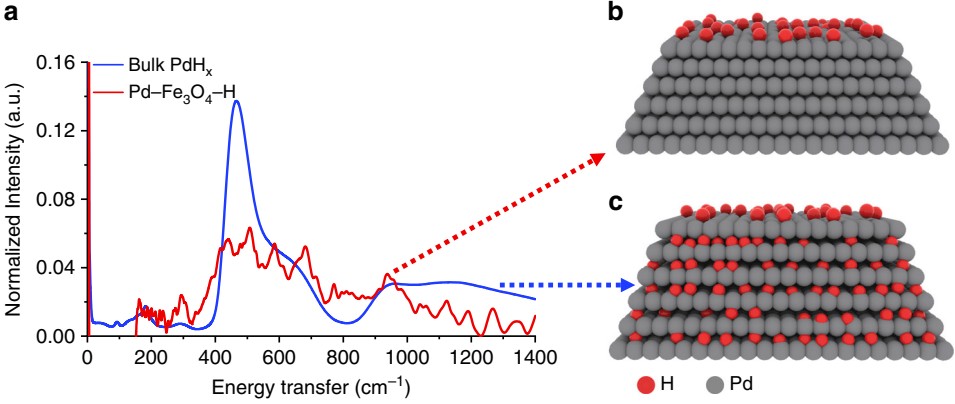

**Fig. 6 Determination of reaction intermediate species over Pd–Fe$_3$O$_4$–H and bulk PdH$_x$. a** INS spectra of Pd–Fe$_3$O$_4$–H and bulk PdH$_x$; **b** schematic diagram of surface-H formed in Pd–Fe$_3$O$_4$–H; and **c** schematic diagram of hydride formed in bulk PdH$_x$. Source data are provided as a Source data file.

**Preparation of supported NPs**. We employed a common-used insert material, $\gamma$-Al$_2$O$_3$, as the support to anchor the Pd–FeO$_x$ NPs (or Fe$_3$O$_4$ NPs, or Pd NPs). Typically, 200 mg of $\gamma$-Al$_2$O$_3$ was dispersed in a mixture of 15 mL of hexane and 20 mL of ethanol under sonication. A total of 20 mg of prepared NPs in 5 mL of hexane was added into the solution dropwise under sonication. The final mixture was further sonicated for 2 h and then magnetically stirred overnight. Subsequently, the NPs/Al$_2$O$_3$ was separated by centrifugation at 6082 × g for 10 min, and washed by 20 mL of ethanol and hexane for two times. The final sample was dried at 50 °C under vacuum overnight.

**Preparation of Pd–Fe$_3$O$_4$–H and Pd–Fe$_3$O$_4$–A**. The Pd–Fe$_3$O$_4$–H sample was prepared as follows: the Pd–FeO$_x$ NPs/Al$_2$O$_3$ (ST2 NPs/Al$_2$O$_3$) was placed in a tube furnace and then heated to 300 °C at a ramping rate of 20 °C min$^{-1}$ and kept there for 1 h under the atmosphere of 4% H$_2$ in Ar atmosphere. The sample obtained was denoted as Pd–Fe$_3$O$_4$–H. The Pd amount was determined to be 0.171 wt.% by ICP. The Pd–Fe$_3$O$_4$–A sample was prepared by treating the Pd–FeO$_x$ NPs/Al$_2$O$_3$ (ST2 NPs/Al$_2$O$_3$) in air under the same reaction condition.

**Preparation of Pd–Fe$_3$O$_4$–Re**. The Pd–Fe$_3$O$_4$–Re sample was prepared by treating Pd–Fe$_3$O$_4$–H in the air for another 1 h.

**Preparation of Fe$_3$O$_4$–H**. The Fe$_3$O$_4$–H sample was prepared by the same process of preparation of Pd–Fe$_3$O$_4$–H by using FeO$_x$ NPs/Al$_2$O$_3$ instead of Pd–FeO$_x$ NPs/Al$_2$O$_3$ .

**Preparation of control samples with different STs**. Control samples with different ST were obtained by a similar synthesis process of Pd–FeO$_x$ NPs using different amounts of Pd NPs seeds (37.5, 12.5, and 6.25 mg), the samples were respectively denoted as ST1 NPs, ST2 NPs, and ST3 NPs. The NPs were then deposited on the $\gamma$-Al$_2$O$_3$ and further treated in the atmosphere of 4% H$_2$ in Ar for 1 h according to the same process of Pd–Fe$_3$O$_4$–H. (ST2 is the Pd–Fe$_3$O$_4$–H sample in this work).

**Preparation of control samples with different structures**. The control samples with different structures were prepared according to a similar process of Pd–Fe$_3$O$_4$–H at different annealing temperatures. The annealing temperature was 200 °C, 300 °C, and 400 °C, and the corresponding samples were denoted as T200, T300, and T400 (T300 is the Pd–Fe$_3$O$_4$–H sample in this work).

**Characterization**. The powder X-ray diffraction (XRD) patterns were collected on a PANalytical X'Pert Pro MPD diffractometer using an X'Celerator RTMS detector. HAADF-STEM and HR-STEM were performed on a Nion Ultra STEM 100 (operated at 100 kV). EELS spectra were collected on a high-resolution Gatan-Enfina ER with a probe size of 1.3 Å. TEM and high-angle annular bright-field scanning transmission electron microscopy (HAABF-STEM) were obtained on a Hitachi HD-200 with bright-field STEM detector operating at 200 kV.

The dispersion of the Pd was evaluated via pulse H$_2$-Chemisorption with an Altamira Instruments (AMI-300) system. Before the measurements, ~100 mg catalyst was pretreated at 550 °C for 3 h under 50 sccm of Ar, followed by cooling down to desired temperature (i.e., −130 and 35 °C) under the same flow. Then pulses of 4% H$_2$/Ar from a sample loop with a defined volume (~0.5 cc) were injected by switching a six-way valve until the eluted peak area of consecutive pulses was constant. The dispersion of Pd was calculated from the volume of H$_2$.

INS experiments were performed at the VISION beamline of the Spallation Neutron Source, Oak Ridge National Laboratory. The Pd–Fe$_3$O$_4$–H sample was first treated under vacuum at 600 °C for 12 h. It was then loaded in an aluminum

sample holder in a helium glovebox. The sample holder was attached to a gas-loading sample stick connected to a gas panel. The blank sample was first measured at −268 °C for 3 h to collect baseline spectrum. H$_2$ gas was then introduced in situ at −238 °C, followed by heating of the sample to −98 °C for reaction. The system was then cooled back to −268 °C to measure the reacted spectrum. The difference spectrum (reacted minus baseline) shows the signal associated with the hydride species formed during the reaction. The CO DRIFTS results were obtained on a Nicolet 670 Fourier Transform Infrared Spectrometer with an MCT detector by the following process: each sample (~15 mg) was loaded and then pretreated at 200 °C under Ar for 30 min. Afterward, the sample was cooled down to −120 °C to conduct CO adsorption. When the temperature reached −120 °C, the background was measured and then CO adsorption was conducted for 30 min as followed by desorption with Ar for 10 min (CO desorbed within 1 min after flow Ar). XPS characterization was performed on a PHI VersaProbe III scanning XPS microscope using a monochromatic Al K-alpha X-ray source (1486.6 eV). XPS spectra were acquired with 200 μm/50 W/15 kV X-ray settings and dual-beam charge neutralization. All binding energies were referenced to Al 2p peak at 74.8 eV.

**Catalytic performance tests**. The hydrogenation of C$_2$H$_2$ was carried out in a tubular quartz reactor with a 0.25-inch diameter. In a typical run, ~15 mg of catalyst was mixed with 150 mg of 60–80 mesh quartz sand and placed in the center of the reactor. The catalyst bed was held by quartz wool at both ends and the reactor was loaded in a vertical furnace (Carbolite Gero). The catalyst was purged with He for 30 min at a flow rate of 20 sccm prior to the reaction under room temperature. Then, the reactor was heated to the desired temperature (i.e., 30–80 °C), followed by feeding the gas mixture (i.e., 0.6 sccm C$_2$H$_2$, 3 sccm H$_2$ balanced with He) at a total flow rate of 50 sccm. The exit gas mixture was analyzed on-line by a ThermoStar Mass Spectrometry (Pfeiffer).

The conversion and selectivity were calculated by using Eqs. (1) and (2):

$$C_2H_2 \; Conversion(\%) = \left(1 - \frac{X_{C_2H_2,out}}{X_{C_2H_2,in}}\right) \times 100\% \qquad (1)$$

$$Selectivity(\%) = \frac{X_{C_2H_4,out}}{X_{C_2H_2,in} - X_{C_2H_2,out}} \times 100\% \qquad (2)$$

whereas in/out refers to the concentration measured in the inlet/outlet port.

Reaction orders with respect to H$_2$ and C$_2$H$_2$ were calculated by the differential method. The corresponding conversion is maintained below 20% to ensure a true kinetic regime. Apparent activation energy is calculated by the Arrhenius equation.

**DFT calculation**. The density functional theory calculations were performed with the Vienna Ab Initio Simulation Package (VASP)[49,50]. The on-site Coulomb interaction was included with the DFT + U method by Dudarev et al.[51] in VASP using a Hubbard parameter U = 3.8 eV for the Fe atom. The Perdew–Burke–Ernzerhof[52] functional form of generalized-gradient approximation was used for electron exchange and correlation energies. The projector augmented-wave method was used to describe the electron–core interaction[49,53]. A kinetic energy cutoff of 450 eV was used for the plane waves. A 3 × 2 × 1 sampling of Brillouin zone using a Monkhorst-Pack scheme was used[54]. A vacuum layer of 15 Å was added for the surface slabs along the z-direction; the slab contains a total of four layers, with the bottom two layers fixed in their bulk positions.

**XAFS data collection and processing**. Approximately 20 mg of sample was enclosed in a nylon washer of 4.953 nm inner diameter and sealed on one side with transparent "Scotch" tape. The sample was pressed by hand to form a uniform pallet, then sealed on the open side with a tape. XAFS investigation were performed

at beamline 10ID-B of the Advanced Photon Source at Argonne National laboratory[55]. Spectra were collected at the iron K-edge (7112 eV) and palladium K-edge (24,350 eV) in transmission mode, with an iron and palladium foil as a reference for energy calibration, respectively. All spectra were collected at room temperature and ten scans were collected for each sample. All data were processed and analyzed using the Athena and Artemis program of the IFFEFFIT package[56] based on FEFF 6.0. Reference foil data were aligned to the first zero-crossing of the second derivative of the normalized $\mu(E)$ data, which was subsequently calibrated to the literature $E_0$ for each Fe K-edge and Pd K-edge. The background was removed, and the data were assigned a Rbkg value of 1.0 prior to normalizing to obtain a unit edge step. All data were initially fit with k-weighting of 1, 2, and 3 then finalized with $k^3$-weighting in R-space. A fit of the Pd foil and Fe foil was used to determine $S_0^2$ for each sample. Structure models used to fit the data sets were obtained from crystal structure of iron oxide and DFT calculation. Structure parameters that were determined by the fits include the degeneracy of the scattering path ($N_{degen}$), the change in Reff, the mean square relative displacement of the scattering element($\sigma_i^2$), and the energy shift of the photoelectron($\Delta E_0$). $k^3$-weighting in R-space. Initial fitting was conducted using crystal structure from crystal database. The simulated models were obtained from DFT calculation and scattering paths of selected scattered atom (Fe, Pd) were generated through FEFF calculation. The WT method was adapted for a quantitative analysis of the back-scattering atom in the higher coordination shells with EvAX code[57].

## Data availability
The data that support the plots within this paper and other findings of this study are available from the corresponding author upon reasonable request. Source data are provided with this paper.

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

## Acknowledgements

This research is sponsored by the U.S. Department of Energy (DOE), Office of Science, Office of Basic Energy Sciences, Chemical Sciences, Geosciences, and Biosciences Division, Catalysis Science Program. The computational calculations used resources of the National Energy Research Scientific Computing Center, a DOE Office of Science User Facility. XAFS data were collected at the Advanced Photon Source at Argonne National Laboratory on Beamline 10ID-B, supported by the Materials Research Collaborative Access Team (MRCAT). MRCAT operations are supported by the DOE and the MRCAT member institutions. This research used resources of the Advanced Photon Source, a U.S. DOE Office of Science User Facility operated for the DOE Office of Science by Argonne National Laboratory under contract no. DE-AC02-06CH11357. The neutron studies used resources at the Spallation Neutron Source, a DOE Office of Science User Facility operated by Oak Ridge National Laboratory. Part of the work including the chemisorption was conducted at the Center for Nanophase Materials Sciences, which is a DOE Office of Science User Facility. The Spallation Neutron Source at Oak Ridge National Laboratory is supported by the Scientific User Facilities Division, Office of Basic Energy Sciences, U.S. DOE, under contract no. DE-AC0500OR22725 with UT Battelle, LLC. Part of the TEM work was performed at the Center for Functional Nanomaterials, Brookhaven National Laboratory, which is supported by the U.S. DOE, Office of Basic Energy Science, under contract no. DE-SC0012704. P.W.W., W.S.Z., and H.M.L. were financially supported by the National Natural Science Foundation of China (21722604), Natural Science Foundation of Jiangsu Province (BK20190852). P.W.W. is thankful to the scholarship from China Scholarship Council (CSC).

## Author contributions

H.Y.Z., P.W.W., S.T., and S.D. conceived the idea of the work. P.W.W. synthesized the samples and carried out the XRD analysis. S.T. performed the catalytic experiments. J.M. and C.W.A. performed the XAFS. J.M., V.F., D.E.J., P.W.W., H.Y.Z., and C.W.A. analyzed the XAFS result, and carried out the DFT simulation. J.M. performed the CO DRIFTS. P.W.W. and H.Y.Z. analyzed the CO DRIFTS results. N.L., D.S., and S.Z.Y. performed the part of the TEM, HAADF-STEM, STEM, HR-STEM, and EELS mapping. P.W.W. and H.Y.Z. performed some of the TEM characterizations. N.L., P.W.W., W.S.Z., Z.H.Y., and H.Y.Z. analyzed the microscopic results. Y.Q.C. and Z.L.W. carried out the INS characterization, and analyzed the results. S.T. performed H₂-chemisorption characterization. P.W.W., S.T., A.S., A.M.M., H.M.L., Z.H.Y., W.S.Z., S.D., and H.Y.Z. discussed the results. Z.H.Y., P.W.W., and H.Y.Z. analyzed the XPS results. P.W.W., S.T., and H.Y.Z. summarized the results, and drafted the manuscript. All authors modified the manuscript. P.W.W., S.T., and H.Y.Z. finalized the manuscript.

## Competing interests

The authors declare no competing interests.
