## [Peer Review File · Nature Communications]

Reviewers' comments:

Reviewer #1 (Remarks to the Author):

This manuscript describes a novel method for construction of strong metal-support interactions (SMSI) by reducing a core-shell system. As a typical run, authors showed the preparation of Pd-FeOx-H by reduction of core-shell Pd@Fe₃O₄, which is very active for semi-hydrogenation of acetylene with 100% conversion and 85.1% selectivity to ethylene at 80 °C. These results are interesting, therefore I recommend to publish this work after revisions in the following:

1. In the part of introduction, authors emphasized the challenge for construction of SMSI for Au with low work function or surface energy. In fact, as reference of 23 cited by authors, it has been reported a construction of SMSI for Au at room temperature, which is very mild. Authors should mention these works in the introduction part, and authors offers an alternative method to construct additional SMSI phenomenon by new route.
2. For the construction of yolk-shell structure, authors should give pore size distribution in the shell structure.
3. For the transformation from core-shell into yolk-shell structure, authors should give the change in pore volume.
4. After deactivation, is it possible to regenerate the activity?

Reviewer #2 (Remarks to the Author):

The authors present an approach to engineer strong metal-support interaction via a reverse route (SMSIR), which increase the catalytic performance in semi-hydrogenation of acetylene. But the manuscript has many questions need to be addressed before it is suitable for publication.

1. The formation process of classical SMSI is reversible. However, the authors does not mention this point, whether SMSIR is reversible or not.
2. Electron transfer between metal and support is very important in SMSI, however, which is almost not investigated in this work.
3. It was reported that the shell thickness in Pd-Fe₃O₄-H decreased with the reduction temperature increased. I am concerned about the morphology of the catalysts when the reduction temperature is higher than 500 oC, whether encapsulation will occur or not.
4. The strategy that changes the structure of the catalyst from complete encapsulation core-shell to yolk-shell with numerous voids looks like some corrosion, which almost has not relationship with SMSI.
5. Catalytic performance stability of Pd-Fe₃O₄-H should be investigated in the model reaction of semi-hydrogenation of acetylene.
6. Why the formation of hydride can be effectively inhibited in Pd-Fe₃O₄-H catalyst with SMSIR? The

PdHx hydride should also be selected as one of the control samples.

Reviewer #3 (Remarks to the Author):

Authors report an interesting route to exploit the well-studied and well reported SMSI feature in Pd NPs supported on Fe₂O₃. This is an interesting article, however I could not see the novelty of this work. For example, back in 2011 Ferdi Schuth (and co workers reported a very similar approach (albeit Au nanoparticles and the reaction used was CO oxidation). However the strategy is similar, if not the same. Hence the novelty of this work should be highlighted in comparison to Schuth's work.

Authors have tuned the SMSI by changing the reduction temperature. This is interesting, however while changing the reduction temperature there is always the possibility for the Pd nanoparticles to grow and this can affect the catalytic activity. What is the particle size distribution of Pd particles for all the reaction temperatures?

The exposed Pd metal surface area should be quantified by CO chemisorption, this data for all the samples would help to prove the hypothesis presented in this article.

Overall it is an interesting concept, however more evidences are needed to prove the claims made in this article.

Incl: Our answers to reviewer's questions

Reviewer #1

Comments: This manuscript describes a novel method for construction of strong metal-support interactions (SMSI) by reducing a core-shell system. As a typical run, authors showed the preparation of Pd-FeOx-H by reduction of core-shell Pd@Fe₃O₄, which is very active for semi-hydrogenation of acetylene with 100% conversion and 85.1% selectivity to ethylene at 80 °C. These results are interesting, therefore I recommend to publish this work after revisions in the following:

Response: We thank the reviewer for his/her recommendation and constructive comments. We have addressing the raised comments point by point as follows:

Q1: In the part of introduction, authors emphasized the challenge for construction of SMSI for Au with low work function or surface energy. In fact, as reference of 23 cited by authors, it has been reported a construction of SMSI for Au at room temperature, which is very mild. Authors should mention these works in the introduction part, and authors offers an alternative method to construct additional SMSI phenomenon by new route.

R1: Thanks for the suggestion. Since the very first discovery of SMSI by Tauster et al., it has been long regarded that construction of SMSI for Au is challenging. Only till very recently, significant progress has been achieved for engineering SMSI on Au surface, including the cited Ref. 23. These discoveries inspired researchers to develop a series of highly-active catalysts. As you mentioned, our work aims to provide a novel route for construction of SMSI to promote catalysis. In the revised manuscript, we have discussed the Ref. 23 according to your suggestion as follows:

“... One critical element in this pursuit is switching the high-temperature treatment in H₂ into other conditions and thereby changes the mechanistic pathways for the formation of SMSI. For example, Xiao et al. reported a novel wet-chemistry approach to constructing SMSI in aqueous solution at room temperature, which was realized by engineering redox interactions between metals and supports. Such strategy is applicable to Au, Pt, Pd, and Rh²³. Christopher et al. developed a strongly-bounded-adsorbate-mediated strategy to construct SMSI between Rh and TiO₂ through high-temperature treatment in the mixture of CO₂ and H₂¹. Zhang et al. engineered the SMSI between Au NPs and hydroxyapatite by treating the Au NP-hydroxyapatite composite in the air at high temperatures². Although some progresses have been made in expanding the boundaries of SMSI, one inevitable issue associated with the conventional SMSI is that upon high-temperature treatment the encapsulation process, immediately and uncontrollably takes place, resulting in limited exposure of active sites³. In the ideal scenario, the oxide coverage on metal surface needs to be thin and permeable to small molecules while still fully encapsulating metal NPs to prevent the dissolution, disintegration, and aggregation of active sites during catalysis...”

Q2: For the construction of yolk-shell structure, authors should give pore size distribution in the shell structure.

R2: Thank you for your suggestion. Since the nanoparticles were anchored on gamma-Al₂O₃, it is unable to determine the pore distribution via N₂ adsorption-desorption curves because of the major contribution of pore size/distribution will come from the Al₂O₃. In order to estimate the pore size, we analyzed seven HR-STEM images, which contain 64 pores in total, to provide the pore distribution. As shown in the results, the pores on the shell are mainly micropores and the majority of the them are < 1 nm. The average pore size was calculated to be 0.73 nm (Fig. A1, Supplementary Fig. 5 in the revised version). The related result has been added to the revised manuscript as follows.

Fig. A1. (Supplementary Fig 5) a-g) HR-STEM images of Pd-Fe₃O₄-H and h) pore size distribution.

“...More interestingly, the magnified HR-STEM image in Fig. 2d shows that there are abundant voids, i.e., lattice vacancies, in the Fe₃O₄ shells (marked in yellow circles). To analyze the pore distribution, the pore sizes were determined by averaging pore sizes in multiple HR-STEM images in Supplementary Fig. 5. The majority of the pores on the Fe₃O₄ shells are micropores with an average size of 0.73 nm.”

Q3: For the transformation from core-shell into yolk-shell structure, authors should give the change in pore volume.

R3: Thank you for the comment. As mentioned above, since these NPs are anchored onto γ-Al₂O₃, it is difficult to determine their pore volume. Here we try to calculate the pore volume via HR-STEM. However, we need mention that since the HR-STEM images are 2D images, it is unable to calculate the volume accurately. After a careful comparison, we found that the sizes of the NPs increased with the formation of yolk-shell structure. Since there is no pore in the pristine core-shell structure, we can assume that the expanded

volume is approximately similar to the pore volume of the yolk-shell structure. The average particle size of the core-shell NPs is ~14 nm. After the transformation from core-shell to yolk-shell structure in H₂ at 300 °C, the average particle size increases to ~17 nm. Therefore, the pore volume of a particle is calculated to ~6810 nm³ according to the equation: $pore\ volume = \pi * R_{Pd-Fe_3O_4-H}^3 - \pi * R_{Pd-FeO_x\ NPs}^3$. However, we didn't choose to add this data into the revised manuscript because the value was only an estimation from the 2D HR-STEM result.

Q4: After deactivation, is it possible to regenerate the activity?

R4: Thanks for the question. Our catalyst didn't deactivate after prolonged time-on-stream tests. To test the stability of the catalyst, we performed the time-on-stream testing to monitor the performance of the catalyst both at the high conversion and the low conversion. As shown in the results that the catalyst possesses excellent stability at both conditions, and no significant deactivation was detected even after 64 h reaction. Most importantly, since the reaction temperature was as low as 80 °C and the catalyst was prepared at 300 °C, we envision the catalyst is very stable at the reaction temperature of 80 °C.

Fig. A2 a) Stability of Pd-Fe₃O₄-H under high conversion; b) Stability of Pd-Fe₃O₄-H under low conversion. Reaction conditions: *m* (catal.) = 15 mg; *v* (gas) = 50 sccm (0.6 sccm C₂H₂, 3 sccm H₂, 46.4 sccm He); *T* = 80 °C.

Reviewer #2

Comments: The authors present an approach to engineer strong metal-support interaction via a reverse route (SMSIR), which increase the catalytic performance in semi-hydrogenation of acetylene. But the manuscript has many questions need to be addressed before it is suitable for publication.

Response: We thank the reviewer for his/her recommendation and constructive comments. We have addressing the raised comments point by point as follows:

Q1: The formation process of classical SMSI is reversible. However, the authors does not mention this point, whether SMSIR is reversible or not.

R1: Thank you for the constructive comment. We have run additional experiments and found that the SMSIR in this work is partially reversible which is slightly different from traditional SMSI. Specifically, in order to determine whether the SMSIR is reversible, we retreated the Pd-Fe₃O₄-H sample in air at 300 °C (denoted as Pd-Fe₃O₄-Re) and corresponding TEM, CO DRIFTS and XPS analysis were performed. Both the TEM, CO DRIFTS result and XPS analysis demonstrate that the SMSIR is partially reversible. The corresponding discussion has been added to the revised manuscript:

“...The Pd-Fe₃O₄-H sample was further retreated in air at 300°C to obtain Pd-Fe₃O₄-Re sample, and characterized by TEM, CO DRIFTS and XPS analysis to determine the reversibility of SMSIR. As shown in the TEM image of the Pd-Fe₃O₄-Re (Fig. A3, Supplementary Fig. 14 in revised manuscript), the sample still possesses a yolk-shell structure, but the voids are smaller than those of Pd-Fe₃O₄-H. The XPS of Pd-Fe₃O₄-Re (Fig. A4, Supplementary Fig. 15 in revised manuscript) demonstrates three Pd states of metallic Pd, Pd^{δ+} in Pd-Fe bond, and PdO. The intensity of Pd^{δ+} peak is lower than that of Pd-Fe₃O₄-H (Fig. A6, Supplementary Fig. 12 in revised manuscript), indicating the decrease of Pd^{δ+} concentration. Besides, it can be seen from the CO DRIFTS of Pd-Fe₃O₄-Re in Fig. A5 (Supplementary Fig. 16 in revised manuscript) that the intensity of CO-Pd^{δ+} peak became weaker than that of Pd-Fe₃O₄-H sample in Fig. A7 (Supplementary Fig. 13 in revised manuscript), suggesting the CO DRIFTS spectral feature is an intermediate state between Pd-Fe₃O₄-H and Pd-Fe₃O₄-A. The analysis of TEM, CO DRIFT and XPS together suggests that the SMSIR in this work is partially reversible...”

Fig. A3 (Supplementary Fig. 14) TEM image of Pd-Fe₃O₄-Re.

Fig. A4 (Supplementary Fig. 15) XPS spectra of Pd-Fe₃O₄-Re.

Fig. A5 (Supplementary Fig. 16) CO DRIFTS of Pd-Fe₃O₄-Re.

Q2: Electron transfer between metal and support is very important in SMSI, however, which is almost not investigated in this work.

R2: Thanks for the constructive comment. We totally agree with you that the electron transfer between the metal and support is very important. Actually, we highlighted the new bond formation between Pd and Fe in the SMSIR, reflecting the electron transfer process. Additionally, in the revised version, we further carried out and analyzed CO DRIFTS and XPS to analyze the electron transfer between Pd and Fe₃O₄. It can be seen from the results that with the formation of SMSIR, electron partially transfers from Pd core to Fe₃O₄ shell, which can be attributed to the intercalation of Pd into Fe₃O₄ shell to form new Pd-Fe bonds, consistent with our WT EXAFS data. Detailed results and related discussion have been added to the revised manuscript as follows:

Fig. A6 (Supplementary Fig. 12) XPS spectra of Pd-FeO_x NPs, Pd-Fe₃O₄-H and Pd-Fe₃O₄-A.

Fig. A7 CO DRIFTS spectra of Pd-Fe₃O₄-H and Pd-Fe₃O₄-A.

“To investigate the charge transfer with the formation of SMSIR, XPS spectra and CO

diffuse reflectance infrared Fourier transform spectroscopy (DRIFTS) were carried out and shown in Figs. A6 and A7 (Supplementary Figs. 12 and 13 in revised manuscript). In the XPS of Pd-FeO_x NPs, only a Pd 3d5/2 peak at 335.4 eV, being assigned to metallic Pd, was found^{4,5}. When the Pd-FeO_x NPs were treated in air at 300 °C, an additional Pd 3d5/2 peak at 336.8 eV assigned to PdO, emerged in Fig. A6 (Supplementary Fig. 12 in revised manuscript). This observation is consistent with our XAFS results that the Pd in Pd-Fe₃O₄-A possesses an oxidized feature to some extent. Furthermore, a Pd 3d5/2 shoulder peak at 336.2 eV in the XPS of Pd-Fe₃O₄-H was detected and assigned to the positively-charged Pd (Pd^{δ+})⁶, which is originated from the intercalation of Pd into Fe₃O₄ matrix, leading to the strong interactions between Pd and Fe to form Pd-Fe bond⁷, in accordance with the XAFS results. The XPS indicates the formation of strong interactions and partial electron transfer between Pd and Fe₃O₄. The CO DRIFTS was further carried out. During the test, we found that the CO adsorption peak was weak, and therefore the pure gas phase signal was subtracted from each data set. As shown in Fig. A7 (Supplementary Fig. 13 in revised manuscript), a peak at ~2153 cm⁻¹ was detected in both the CO DRIFTS spectra of Pd-Fe₃O₄-H and Pd-Fe₃O₄-A, which is assigned to Fe³⁺-CO⁸. Due to the core-shell morphology of Pd-Fe₃O₄-A, the Pd core is fully encapsulated by Fe₃O₄ shell, no obvious peak of CO-Pd was detected in the CO DRIFTS of Pd-Fe₃O₄-A. In contrast, a very weak peak at 2102 cm⁻¹ can be seen in the CO DRIFTS of Pd-Fe₃O₄-H, which is assigned to the linear-CO adsorption on metallic Pd⁹. More interestingly, an additional peak at 2134 cm⁻¹ can be found. Compared with the position of linear-CO adsorption on metallic Pd, the peak blueshifted, being assigned to linear-CO adsorption on positively-charged Pd (CO-Pd^{δ+})¹⁰. Combined with the XPS analysis, this peak may be assigned to linear-CO adsorption on Pd^{δ+} in Pd-Fe bond, indicating that electron transfers from Pd to Fe₃O₄.”

Q3: It was reported that the shell thickness in Pd-Fe₃O₄-H decreased with the reduction temperature increased. I am concerned about the morphology of the catalysts when the reduction temperature is higher than 500 oC, whether encapsulation will occur or not.

R3: Thank you for the question. In this work, we treated the started Pd-FeO_x NPs at different temperatures (from 200 to 400 °C). As shown in Fig. 1, Supplementary Fig. 34 and 35, for different samples obtained at different temperatures, the morphologies vary significantly. For T200 sample, because of the low treating temperature, the STEM showed that the T200 sample is a core-shell structure and only few voids were detected in the shell. With the treating temperature increased to 300 °C, the STEM characterization of Pd-Fe₃O₄-H (T2) in Fig. 2 illustrates that the NPs transformed from a pristine core-shell structure to final yolk-shell structure with numerous voids in the shell. Moreover, the XANES results based on DFT calculation demonstrate that the Pd atoms are partly intercalated into Fe₃O₄ shell, indicating the formation of strong interactions between Pd and the Fe₃O₄. When the treating temperature increased to 400 °C, the

core-shell structure was totally destroyed, and the structure became Pd islands on Fe_3O_4 . If the treating temperature is $> 500\text{ }^\circ\text{C}$, we expect that the Fe_3O_4 would be reduced to metallic Fe. And with the formation of metallic Fe, the Fe tends to alloy with Pd to form PdFe alloy. In previous work by Hou et al.¹¹, they treated the Pd- FeO_x core-shell nanoparticles at $500\text{ }^\circ\text{C}$ in H_2/Ar , and a PdFe alloy/ α -Fe nanocomposite was obtained. As a result, it can be speculated that if the treating temperature is $> 500\text{ }^\circ\text{C}$, the core-shell structure Pd- FeO_x NPs would transform to PdFe alloy and metallic Fe and the encapsulation will no longer exist.

Q4: The strategy that changes the structure of the catalyst from complete encapsulation core-shell to yolk-shell with numerous voids looks like some corrosion, which almost has not relationship with SMSI.

R4: Thank you for your comments. We agree with you that there is indeed some similarity between corrosion and the transformation from core-shell to yolk-shell when the Pd- FeO_x NPs were treated in H_2 at $300\text{ }^\circ\text{C}$. The corrosion is originated from the partially reduction of Fe_3O_4 shell in H_2 . However, we would like to emphasize that the concept of SMSI is related into two main aspects: 1) encapsulation which in our case is the encapsulation of Pd by Fe_3O_4 shell; even though the hydrogen treated one demonstrated an intermediate state of SMSI, it still presents the feature of SMSI (the coverage of metal by metal oxide). This intermediate SMSI state actually benefits the following catalytic study because it can provide much more abundant active sites than conventional SMSI with complete encapsulation; 2) the electron transfer between metal and metal oxide which was proved by our XAFS, XPS, and CO DRIFTS investigations. In conclusion, although the fabrication process does look like a corrosion process, the final structure features important characteristics of SMSI.

Q5: Catalytic performance stability of Pd- Fe_3O_4 -H should be investigated in the model reaction of semi-hydrogenation of acetylene.

R5: Thank you for your question. To test the stability of the catalyst, we performed the prolonged time-on-stream tests to monitor the performance of the catalyst both at high conversion and low conversion. As shown in the results (Fig. A2), the catalyst possesses excellent stability both at high conversion and low conversion, and no significant deactivation was detected. The stability experiments illustrate that the Pd- Fe_3O_4 -H catalyst possesses an excellent catalytic performance stability.

Fig. A2 a) Stability of Pd-Fe₃O₄-H under high conversion; b) Stability of Pd-Fe₃O₄-H under low conversion. Reaction conditions: m (catal.) = 15 mg; v (gas) = 50 sccm (0.6 sccm C₂H₂, 3 sccm H₂, 46.4 sccm He); T = 80 °C.

Q6: Why the formation of hydride can be effectively inhibited in Pd-Fe₃O₄-H catalyst with SMSIR? The PdH_x hydride should also be selected as one of the control samples.

R6: Thanks for the question. Since the response to the first question is very long, we would like to answer the second question first. In our experiment, we employed Pd as a control sample. Based on previous studies, it was found that Pd can readily react with hydrogen to form PdH_x hydride at room temperature^{12,13}. For example, Whitlam et al. reported the formation of PdH_x hydride by exposing Pd in H₂. Similarly, during our catalytic hydrogenation process, the Pd NPs reacted with H₂ to form PdH_x hydride, and further reacted with C₂H₂. Undoubtedly, if pure PdH_x hydride is employed in acetylene hydrogenation, the catalytic performance will be the same as the Pd NPs (100% of conversion and 0% of selectivity).

For the second question, we found the hydrogen adsorption on Pd-Fe₃O₄ is much weaker than that on pure Pd (Supplementary Fig. 39). In addition, previously, Albers et al.¹⁴ reported that alloying Pd with Fe can reduce the formation of hydride, which is the main source for total hydrogenation. In our system we found that with the formation of SMSIR, the Pd was slightly positively charged because of the formation of Pd-Fe bond (Supplementary Figs. 12 and 13), which could also contribute to the inhibition of formation of hydride according to previous reports¹⁴.

Furthermore, we carried out theoretical calculation to show that why such a structure in this work can inhibit the formation of hydride (details of calculation in the Appendix). We evaluated the possibility that the iron oxide shell prevents the Pd core from absorbing

hydrogen into a subsurface state, i.e., forming hydride. A large quantity of literature is available for the energetics and kinetics of hydrogen adsorption and absorption into Pd particles.¹⁵⁻⁶⁶ The available information was used to approximate the ratios of surface in subsurface hydrogen expected under these conditions (the details used for the calculation are provided in the Appendix). 5 nm Pd particles are expected to have significant subsurface hydrogen, in the absence of any modulating factors. The surface to subsurface hydrogen ratio is expected to be on the order of 0.1:1 to 0.8:1 for this size particle; which means more subsurface hydrogen than surface hydrogen. This is consistent with the neutron data for the pure Pd particles. The formation of palladium hydride is accompanied by a volume expansion.²³ The subsurface hydrogen is expected to be > 80% filled under these conditions, which corresponds to a volume expansion of >10%.²³

In contrast, our experimental data shows that the iron oxide encapsulated Pd particles do not absorb hydrogen to create subsurface hydrogen. We believe that this change is due to the iron oxide shell restricting the Pd particle such that the volume expansion cannot occur. In effect, the core is sterically hindered from converting into PdH_x. To explore this possibility, we evaluated and answered the following questions based on the material physical properties^{23,67-71} (details shown in the Appendix). **a)** Could the Pd core expand while converting into PdH_x, without cracking the shell? No: the stress created at the interface would be on the order of 10,000 MPa, while the tensile strength of iron oxide is on < 100 MPa. **b)** Could the core convert to PdH_x without expanding? No: the stress caused by the PdH 'wanting' to expand would again be on the order of 10,000 MPa and would crack the shell. Thus, the observations that the shell is still present and that PdH_x is not formed suggest that PdH_x formation is not energetically favored enough to crack the shell or that the process of doing so is very slow. This led to a third question: **c)** Do the thermodynamics of the system favor PdH_x formation and shell cracking? The answer is no. The upper limit for the energy that would be gained by PdH_x formation is on the order of 50 MJ/mol, while the order of magnitude in energy required to crack the shell on the order of 150 MJ/mol: thus, for thermodynamic reasons, the energy that would be gained from PdH_x formation would be insufficient to crack the shell.

In conclusion, the hydride formation is inhibited because of 1) hydrogen adsorption became weaker in Pd-Fe₃O₄-H; 2) Pd-Fe bond formation in Pd-Fe₃O₄-H samples which provides positively charged Pd; 3) sterically hindered expansion of Pd lattice in the encapsulated structure.

Reviewer #3

Comments: Authors report an interesting route to exploit the well-studied and well reported SMSI feature in Pd NPs supported on Fe₂O₃. This is an interesting article, however I could not see the novelty of this work. For example, back in 2011 Ferdi Schuth (and co workers reported a very similar approach (albeit Au nanoparticles and the reaction used was CO oxidation). However the strategy is similar, if not the same. Hence

the novelty of this work should be highlighted in comparison to Schuth's work.

Response: We want to thank the reviewer for his/her time and comments. However, we respectfully disagree with the recommendation from this reviewer and believe there is some level of misunderstanding of our content by the reviewer. We would like to clarify here. Ferdi Schuth and coworkers reported a branch of gold yolk-shell nanoparticles (such as Au@C and Au @ZrO₂) and the nanoparticles were employed in CO oxidation⁷²⁻⁷⁴. These investigations are very interesting and novel. However, our reported strategy in this work is completely different from Schuth's works. First, the preparation process is totally different from Schuth's. In Schuth's reports, they employed an exotemplate method to prepare yolk-shell nanoparticles. For instance, in their reported work (Chem. Eur. J. 2011, 17, 8434-8439; Catal. Sci. Technol., 2011, 1, 65-68), they employed SiO₂ as the exotemplate to fabricate a core-shell structure. Afterward, ZrO₂ was deposited on SiO₂ shell to form a core-shell-shell structure. Then, the Au, @ZrO₂ yolk-shell structure was formed by removing SiO₂ template by NaOH. In this work, the preparation process is totally different. We first prepared a core-shell structured Pd-FeO_x NPs. Afterwards, the core-shell structure transformed to a yolk-shell structure by treating in H₂ at 300 °C. We also found that the core-shell remained when the NPs were treated in air. Second, in this work, we found that strong metal support interactions were formed between Pd core and Fe₃O₄ shell. Schuth's works reported the construction of yolk-shell materials mainly based on Au and ZrO₂ or carbon. In our work, although the final structure for Pd-Fe₃O₄-H is a yolk-shell structure, both the preparation process and materials are totally different. More importantly, in this manuscript, we are focused to design a new strategy to engineer strong metal-support interaction via a reverse route (SMSIR). The construction of SMSIR was realized by starting from the final architecture of SMSI (fully encapsulation, core-shell structure) to obtain the intermediate state with favorable exposure of metal sites. To verify the concept, core-shell Pd-FeO_x NPs were employed as an example. By treating in the reductive atmosphere, the core-shell Pd-FeO_x NPs can be transformed to a unique yolk-shell structure with numerous voids in the shell along with the SMSIR formation. Detailed experiments indicate that the optimized shell thickness and treating condition are essential to the formation of SMSIR. The obtained Pd-Fe₃O₄-H was used for catalytic acetylene semi-hydrogenation because this reaction is sensitive to the structure of Pd. The Pd-Fe₃O₄-H showed 100% of conversion with a selectivity of 85.1% at a mild reaction temperature as low as 80 °C, outperforming most of the previously reported catalysts, including some single-atom catalysts. X-ray spectroscopic investigations coupled with density functional theory simulations and systematic studies using neutron scattering and catalytic evaluations revealed that the outstanding catalytic performance stemmed from SMSIR, enabling the preferred formation of surface-adsorbed hydrogen instead of hydride on Pd. This new strategy of SMSIR allows us to fine-tune the interfacial architecture of supported catalysts and expose the catalytic active sites in a controllable manner.

To the best of our knowledge, this is *the first study* of engineering SMSI via a reverse route, expanding the boundaries of conventional SMSIs and paving the road for a rational design of supported metal catalysts.

Q1: Authors have tuned the SMSI by changing the reduction temperature. This is interesting, however while changing the reduction temperature there is always the possibility for the Pd nanoparticles to grow and this can affect the catalytic activity. What is the particle size distribution of Pd particles for all the reaction temperatures?

R1: Thank you for the question. Indeed, treating temperature has an effect on the size of the NPs, and agglomeration is generally detected when the nanoparticles are treated at high temperature. Fortunately, in this work, we employed core-shell structured nanoparticles as the starting material. With the existence of the iron oxide shell, the Pd NPs can be well protected and prevented from agglomeration. To determine the sizes of Pd particles in different samples, STEM images are provided in Fig. A8 (Supplementary Fig. 38). As shown here, when the Pd-FeO_x NPs were treated at different temperatures, the sizes of Pd particles showed no significant difference. The results illustrate that the sizes of Pd particles in this work are not affected by treating temperatures because of the confining effect of the iron oxide shells. For the size change of Pd particles during the catalytic process, the reaction temperature is very low (< 80 °C), the temperature would not affect the Pd particle sizes significantly.

Fig. A8 (Supplementary Fig. 38) HR-STEM images of Pd-Fe₃O₄ obtained at different temperatures.

Q2: The exposed Pd metal surface area should be quantified by CO chemisorption, this data for all the samples would help to prove the hypothesis presented in this article.

R2: Thank you for the suggestion. CO chemisorption is a powerful tool to determine the metal dispersion and some other characteristics. However, since the iron oxide shell can also adsorb CO molecules to form carbonates, quantification with CO chemisorption is problematic. Consequently, in this work we employed H₂-chemisorption to determine the metal dispersion and particle sizes in supplementary information (Supplementary Table 8). We added CO DRIFTS data for all samples together with XPS results in the revised manuscript. Indeed, the CO adsorption on all of our samples are very weak compared to the CO adsorption on iron species. The detailed information is listed below:

Fig. A6 (Supplementary Fig. 12) XPS spectra of Pd-FeO_x NPs, Pd-Fe₃O₄-H and Pd-Fe₃O₄-A.

Fig. A7 CO DRIFTS spectra of Pd-Fe₃O₄-H and Pd-Fe₃O₄-A.

“To investigate the charge transfer with the formation of SMSIR, XPS spectra and CO diffuse reflectance infrared Fourier transform spectroscopy (DRIFTS) were carried out and shown in Figs. A6 and A7 (Supplementary Figs. 12 and 13 in revised manuscript). In the XPS of Pd-FeO_x NPs, only a Pd 3d_{5/2} peak at 335.4 eV, being assigned to metallic Pd, was found^{4,5}. When the Pd-FeO_x NPs were treated in air at 300 °C, an additional Pd 3d_{5/2} peak at 336.8 eV assigned to PdO, emerged in Fig. A6 (Supplementary Fig. 12 in revised manuscript). This observation is consistent with our XAFS results that the Pd in Pd-Fe₃O₄-A possesses an oxidized feature to some extent. Furthermore, a Pd 3d_{5/2} shoulder peak at 336.2 eV in the XPS of Pd-Fe₃O₄-H was detected and assigned to the positively-charged Pd (Pd^{δ+})⁶, which is originated from the intercalation of Pd into Fe₃O₄ matrix, leading to the strong interactions between Pd and Fe to form Pd-Fe bond⁷, in accordance with the XAFS results. The XPS indicates the formation of strong interactions and partial electron transfer from between Pd and Fe₃O₄. The CO DRIFTS was further carried out. During the test, we found that the CO adsorption peak was weak, and therefore the pure gas phase signal was subtracted from each data set. As shown in Fig. A7 (Supplementary Fig. 13 in revised manuscript), a peak at ~2153 cm⁻¹ was detected in both the CO DRIFTS spectra of Pd-Fe₃O₄-H and Pd-Fe₃O₄-A, which is assigned to Fe³⁺-CO⁸. Due to the core-shell morphology of Pd-Fe₃O₄-A, the Pd core is fully encapsulated by Fe₃O₄ shell, no obvious peak of CO-Pd was detected in the CO DRIFTS of Pd-Fe₃O₄-A. In contrast, a very weak peak at 2102 cm⁻¹ can be seen in the CO DRIFTS of Pd-Fe₃O₄-H, which is assigned to the linear-CO adsorption on metallic Pd⁹. More interestingly, an additional peak at 2134 cm⁻¹ can be found. Compared with the position of linear-CO adsorption on metallic Pd, the peak blueshifted, being assigned to

linear-CO adsorption on positively-charged Pd (CO-Pd^{δ+})¹⁰. Combined with the XPS analysis, this peak may be assigned to linear-CO adsorption on Pd^{δ+} in Pd-Fe bond, indicating that electron transfers from Pd to Fe₃O₄.”

Supplementary Table 8. Dispersion and Pd particle sizes of Pd-Fe₃O₄-H and Pd/Al₂O₃ determined by H₂-chemisorption at different temperatures.

	Pd-Fe ₃ O ₄ -H		Pd	
Temp (°C)	35	- 130	35	- 130
Dispersion (%)	24.4	26.7	72	7.6
Particle size (nm)	4.6	4.2	1.6	14.8
Particle size from STEM (nm)	5.5 ± 0.5		-	

Q3: Overall it is an interesting concept, however more evidences are needed to prove the claims made in this article.

R3: Thank you for the comment. To make the SMSIR concept more convincing, a series of characterizations including XPS, CO DRIFTS, TEM and reversibility of SMSIR were added to the revised manuscript based on the reviewers’ comments, and related discussions were added in the revised manuscript as well. We hope the current version has addressed all your concerns and thank you again for reviewing our manuscript.

Appendix. Detailed calculation process on why the structure of the catalyst can hinder the formation of hydride.

I. Background and Description of Pd-H System

This work concerns calculation of the surface to subsurface hydrogen ratio on Palladium nanoparticles.

The different types of hydrogen species can be classified as

- a) Surface Hydrogen
- b) Immediate Subsurface Hydrogen (or Near Surface Hydrogen)
- c) Bulk-like Subsurface Hydrogen
- d) Bulk Subsurface Hydrogen

The surface hydrogen is the hydrogen that is *adsorbed* directly on the surface, and this hydrogen is generally the most strongly bound of the above-mentioned species. The immediate subsurface hydrogen exists in the first subsurface layer and has different energetics relative to the deeper subsurface hydrogen (in the zero-concentration limit, it is more strongly bound). The bulk like subsurface hydrogen occurs in the second subsurface layer and further, and has energetics that are nearly identical to bulk subsurface hydrogen. During the simulations conducted here, the concentration differences between the second subsurface layer and third subsurface layers were generally minuscule (<1% difference), and the concentration differences between the third and eighth subsurface layer were even smaller. Bulk subsurface hydrogen is deep enough that the different layers have indistinguishable energetics at that depth. In this study, a conservative choice was chosen such that subsurface hydrogen was considered as bulk subsurface hydrogen beginning at the 8th subsurface layer. Based on the concentrations observed in this work, the bulk-like subsurface hydrogen was already indistinguishable from bulk subsurface hydrogen by the 4th subsurface layer, or sooner.

At typical temperatures (< 300 C), the H/Pd phase diagram includes an a phase at extremely low concentrations (typically H/Pd < 0.05), a b phase at high concentrations (typically H/Pd > ~0.6), and a region where the two phases coexist in a tie-line region inbetween.¹⁵⁻¹⁹ The boundaries of the tie line region are temperature dependent, and the region of the phase diagram with the coexistence has a critical temperature of ~565 K and a critical H/Pd ratio of ~0.28.^{16,20} More details can be found in the literature.^{15-18,21,22}

The Pd-H system is well studied with good agreement between different experimental groups for measurements of bulk H/Pd ratios under different temperature and pressure conditions. This enables us to calculate the bulk H/Pd ratio using thermodynamics and equations fitted to existing experimental data. These equations are then extended to the bulk H/Pd ratios inside nanoparticles. The surface

hydrogen energetics and kinetic parameters were taken from the best fits to a combination of experimental and theoretical data.

For Palladium nanoparticles, it is also known that sufficiently small nanoparticles (typically < 6 nm in diameter) exhibit different energetics for H absorption due to “finite-size” energetics, such that the greater pressures are required to achieve the same extent of hydrogen absorption.^{19,23-26,28,29,75} Additionally, the b phase persists to much lower H/Pd ratios before the high-concentration side of the coexistence region is reached, at H/Pd ~0.15. For sufficiently large particles (tens of nm),^{19,26} large-size energetics are achieved which match the measurements of macroscopic Pd. In this work, we considered both cases (large-size energetics and finite-size energetics) for the bulk and bulk-like H species. The large-size energetics were based on bulk material measurements available in the literature, while the finite-size energetics are based on < 6 nm particle measurements and theory. From the knowledge that the nanoparticles used in this work are 10-15 nm in diameter, we anticipated the large-size energetics to be a better match to our experiments, but the finite-size predictions provide another limit that we can compare to, since it’s possible that the 10-15 nm sized particles used here would fall in-between the two limits.

In total, in this work, energetics and kinetic parameters were utilized to calculate the surface to subsurface hydrogen ratio for Palladium nanoparticles, as a function of pressure and temperature. This ratio is then compared to experimental data collected using vibrational neutron spectroscopy.

II. Qualitative Energy Diagram for H on/in Pd

In the limit of low concentration, the potential energy diagram has the form shown in Fig. AS1.³⁰⁻³² This diagram indicates the energetics of electronic states relative to H_2 gas at infinite distance from the surface. The energetics displayed for the adsorbed and absorbed states are approximately equal to the enthalpy of sorption for a single H atom, relative to being in gas phase H_2 .

Fig. AS1. Potential energy diagram for hydrogen adsorption and absorption on and in Pd.

III. Energetics and Kinetic Parameters

Table: Energetics and Kinetic Parameters for a PdH with large-size energetics

Quantity/Variable	Value/Function ^a	References ^a
E_H ₂ (gas)	0 (all state energies are relative to E_H ₂)	N/A
A _{ads}	$0.1 * 1 / \sqrt{2\pi m_{H_2} k_B T}$	33 34
Ea _{ads}	$0 + 54,900 * \theta_{Surf}^2$	33,34 35-37
A _{rec}	10^{13}	33 38
Ea _{rec}	$85772 - 7982.4 * \theta_{Sub1}$	33,34
E _{surf_H} (ads)	$0.5 * (-85772 + 7982.4 * \theta_{Sub1} + Int_{Surf})$	21,33,34,36 31,37,41,43-45,76,77
Int _{Surf}	$54,900 * \theta_{Surf}^2$	34 45
A _{Surf_Sub1}	$4.83 * 10^{13}$	38
Ea _{Surf_Sub1}	$28946 - Int_{Surf} + 0.5 * Int_{Sub1}$	31,32,36,41 37,45-47
A _{Sub1_Surf}	$2.3 * 10^{13} / 3$	38 17,47
Ea _{Sub1_Surf}	$9649 + Int_{Surf} - 0.5 * Int_{Sub1}$	31,41 47
E _{Sub1} (abs)	$-17367 + Int_{Sub1}$	23,31 41,46,48
Int _{Sub1}	$0.5 * (-80200 * \theta_{Sub1}^4 + 175000 * \theta_{Sub1}^3 - 58000 * \theta_{Sub1}^2 - 28800 * \theta_{Sub1}^1)$	19,23 45,46
A _{Sub1_Sub2}	$2.3 * 10^{13} / 3$	38 17,47
Ea _{Sub1_Sub2}	$26051 - 0.5 * Int_{Sub1} + 0.5 * Int_{Sub2}$	31,41 47
A _{Sub2_Sub1}	$2.3 * 10^{13} / 3$	38 17,47
Ea _{Sub2_Sub1}	$19297 - 0.5 * Int_{Sub2} + 0.5 * Int_{Sub1}$	31,41 47
E _{Sub2} (abs)	$-10613 + Int_{Sub2}$	21,30,36,49 18,23,50

Int _{Sub2}	$1*(-80200*\theta_{\text{Sub2}}^4 + 175000*\theta_{\text{Sub2}}^3 - 58000*\theta_{\text{Sub2}}^2 - 28800*\theta_{\text{Sub2}}^1)$	23 17,18,50,51
A_Sub2_Sub3	$2.3*10^{13}/3$	38 17,30,47
Ea_Sub2_Sub3	$23156 - 0.5*\text{Int}_{\text{Sub2}} + 0.5*\text{Int}_{\text{Sub3}}$	17,21,22,30,31,36,41,52-54 47,50,55
A_Sub3_Sub2	$2.3*10^{13}/3$	38 17,30,47
Ea_Sub3_Sub2	$23156 - 0.5*\text{Int}_{\text{Sub3}} + 0.5*\text{Int}_{\text{Sub2}}$	17,21,22,30,31,36,41,52-54 47,50,55
E_Sub3 (abs)	$-10613 + \text{Int}_{\text{Sub3}}$	21,30,36,49 18,23,50
Int _{Sub3}	$1*(-80200*\theta_{\text{Sub3}}^4 + 175000*\theta_{\text{Sub3}}^3 - 58000*\theta_{\text{Sub3}}^2 - 28800*\theta_{\text{Sub3}}^1)$	23 17,18,50,51

^a All state energies are denoted by E_ and are relative to the gas phase 1/2 H₂ and are in kJ/mol. The activation energies are denoted by Ea_ and are in kJ/mol. The interaction terms are denoted by Int_ and are in kJ/mol. The Pre-exponentials are denoted by A_ and are in the appropriate units for the respective rate equations (s⁻¹ for all cases except adsorption).

^a References are provided in the following format: Primary Reference(s) | Supporting Reference(s). The functional forms and values were either directly taken from, based on fitting data from, or found to be accurate enough from the data within the primary reference, while the supporting references are also consistent with the choices made.

Table: Energetics and Kinetic Parameters for b PdH with large-size energetics

Quantity/Variable	Value/Function ^a	References ^a
E_H ₂ (gas)	0 (all state energies are relative to E_H ₂)	33 34
A _{ads}	$0.1 * 1 / \sqrt{2\pi m_{H_2} k_B T}$	33,34 35-37
Ea _{ads}	$0 + 54,900 * \theta_{Surf}^2$	33 38
A _{rec}	10^{13}	33,34
Ea _{rec}	$85772 - 7982.4 * \theta_{Sub1}$	21,33,34,36 31,37,41,43-45,76,77
E _{surf_H} (ads)	$0.5 * (-85772 + 7982.4 * \theta_{Sub1} + Int_{Surf})$	34 45
Int _{Surf}	$54,900 * \theta_{Surf}^2$	38
A _{Surf_Sub1}	$4.83 * 10^{13}$	31,32,36,41 37,45-47
Ea _{Surf_Sub1}	$9649 - (E_{surf_H} - E_{Sub1}) - Int_{Surf} + 0.5 * Int_{Sub1}$	38 17,47
A _{Sub1_Surf}	$2.3 * 10^{13} / 3$	31,41 47
Ea _{Sub1_Surf}	$9649 + Int_{Surf} - 0.5 * Int_{Sub1}$	23,31 41,46,48
E _{Sub1} (abs)	$0.5 (78881 * \theta_{Sub1} - 82231) - 6750$	16 45,46
Int _{Sub1}	$0.5 * (-80200 * \theta_{Sub1}^4 + 175000 * \theta_{Sub1}^3 - 58000 * \theta_{Sub1}^2 - 28800 * \theta_{Sub1}^1)$	38 17,47
A _{Sub1_Sub2}	$2.3 * 10^{13} / 3$	31,41 47
Ea _{Sub1_Sub2}	$26051 - 0.5 * Int_{Sub1} + 0.5 * Int_{Sub2}$	38 17,47
A _{Sub2_Sub1}	$2.3 * 10^{13} / 3$	31,41 47
Ea _{Sub2_Sub1}	$19297 - 0.5 * Int_{Sub2} + 0.5 * Int_{Sub1}$	21,30,36,49 18,23,50
E _{Sub2} (abs)	$0.5 * (78881 * \theta_{Sub1} - 82231)$	16 15

Int _{Sub2}	$1*(-80200*\theta_{\text{Sub2}}^4 + 175000*\theta_{\text{Sub2}}^3 - 58000*\theta_{\text{Sub2}}^2 - 28800*\theta_{\text{Sub2}}^1)$	38 17,30,47
A_Sub2_Sub3	$2.3*10^{13}/3$	17,21,22,30,31,36,41,52-54 47,50,55
Ea_Sub2_Sub3	$23156 - 0.5*\text{Int}_{\text{Sub2}} + 0.5*\text{Int}_{\text{Sub3}}$	38 17,30,47
A_Sub3_Sub2	$2.3*10^{13}/3$	17,21,22,30,31,36,41,52-54 47,50,55
Ea_Sub3_Sub2	$23156 - 0.5*\text{Int}_{\text{Sub3}} + 0.5*\text{Int}_{\text{Sub2}}$	21,30,36,49 18,23,50
E_Sub3 (abs)	$0.5*(78881 *\theta_{\text{Sub1}} - 82231)$	16 15
Int _{Sub3}	$1*(-80200*\theta_{\text{Sub3}}^4 + 175000*\theta_{\text{Sub3}}^3 - 58000*\theta_{\text{Sub3}}^2 - 28800*\theta_{\text{Sub3}}^1)$	33 34

^a All state energies are denoted by E₋ and are relative to the gas phase 1/2 H₂ and are in kJ/mol. The activation energies are denoted by Ea₋ and are in kJ/mol. The interaction terms are denoted by Int₋ and are in kJ/mol. The Pre-exponentials are denoted by A₋ and are in the appropriate units for the respective rate equations (s⁻¹ for all cases except adsorption). For b PdH, the bulk and immediate subsurface hydrogen state concentrations were calculated directly, based on the equations provided in the next section. The energetics listed for the state energies correspond to the enthalpies from those equations.

^a References are provided in the following format: Primary Reference(s) | Supporting Reference(s). The functional forms and values were either directly taken from, based on fitting data from, or shown to be accurate enough from the data within the primary reference, while the supporting references are also consistent with the choices made.

Table: Energetics and Kinetic Parameters for b PdH with finite-size energetics

Quantity/Variable	Value/Function ^a	References ^a
E_H ₂ (gas)	0 (all state energies are relative to E_H ₂)	33 34
A_ads	$0.1 * 1 / \sqrt{2\pi m_{H_2} k_B T}$	33,34 35-37
Ea_ads	$0 + 54,900 * \theta_{Surf}^2$	33 38
A_rec	10^{13}	33,34
Ea_rec	$85772 - 7982.4 * \theta_{Sub1}$	21,33,34,36 31,37,41,43-45,76,77
E_surf_H (ads)	$0.5 * (-85772 + 7982.4 * \theta_{Sub1} + Int_{Surf})$	34 45
Int _{Surf}	$54,900 * \theta_{Surf}^2$	38
A_Surf_Sub1	$4.83 * 10^{13}$	31,32,36,41 37,45-47
Ea_Surf_Sub1	$9649 - (E_{surf_H} - E_{Sub1}) - Int_{Surf} + 0.5 * Int_{Sub1}$	38 17,47
A_Sub1_Surf	$2.3 * 10^{13} / 3$	31,41 47
Ea_Sub1_Surf	$9649 + Int_{Surf} - 0.5 * Int_{Sub1}$	23,31 41,46,48
E_Sub1 (abs)	$0.5 * (78881 * \theta_{Sub1} - 91000) - 6750$	24,29
Int _{Sub1}	$0.5 * (-80200 * \theta_{Sub1}^4 + 175000 * \theta_{Sub1}^3 - 58000 * \theta_{Sub1}^2 - 28800 * \theta_{Sub1})$	38 17,47
A_Sub1_Sub2	$2.3 * 10^{13} / 3$	31,41 47
Ea_Sub1_Sub2	$26051 - 0.5 * Int_{Sub1} + 0.5 * Int_{Sub2}$	38 17,47
A_Sub2_Sub1	$2.3 * 10^{13} / 3$	31,41 47
Ea_Sub2_Sub1	$19297 - 0.5 * Int_{Sub2} + 0.5 * Int_{Sub1}$	21,30,36,49 18,23,50
E_Sub2 (abs)	$0.5 * (78881 * \theta_{Sub1} - 91000)$	24,29

Int _{Sub2}	$1*(-80200*\theta_{\text{Sub2}}^4 + 175000*\theta_{\text{Sub2}}^3 - 58000*\theta_{\text{Sub2}}^2 - 28800*\theta_{\text{Sub2}}^1)$	38 17,30,47
A_Sub2_Sub3	$2.3*10^{13}/3$	17,21,22,30,31,36,41,52-54 47,50,55
Ea_Sub2_Sub3	$23156 - 0.5*\text{Int}_{\text{Sub2}} + 0.5*\text{Int}_{\text{Sub3}}$	38 17,30,47
A_Sub3_Sub2	$2.3*10^{13}/3$	17,21,22,30,31,36,41,52-54 47,50,55
Ea_Sub3_Sub2	$23156 - 0.5*\text{Int}_{\text{Sub3}} + 0.5*\text{Int}_{\text{Sub2}}$	21,30,36,49 18,23,50
E_Sub3 (abs)	$0.5*(78881 *\theta_{\text{Sub1}} - 91000)$	24,29
Int _{Sub3}	$1*(-80200*\theta_{\text{Sub3}}^4 + 175000*\theta_{\text{Sub3}}^3 - 58000*\theta_{\text{Sub3}}^2 - 28800*\theta_{\text{Sub3}}^1)$	33 34

^a All state energies are denoted by E_ and are relative to the gas phase 1/2 H₂ and are in kJ/mol. The activation energies are denoted by Ea_ and are in kJ/mol. The interaction terms are denoted by Int_ and are in kJ/mol. The Pre-exponentials are denoted by A_ and are in the appropriate units for the respective rate equations (s⁻¹ for all cases except adsorption). For b PdH, the bulk and immediate subsurface hydrogen state concentrations were calculated directly, based on the equations provided in the next section. The energetics listed for the state energies correspond to the enthalpies from those equations.

^a References are provided in the following format: Primary Reference(s) | Supporting Reference(s). The functional forms and values were either directly taken from, based on fitting data from, or shown to be accurate enough from the data within the primary reference, while the supporting references are also consistent with the choices made.

IV. Thermodynamic Equations and Other Equations:

The thermodynamic equations are all written for the following standard equilibrium constant:

$$K^{\circ} = (\theta_B^2 / \theta^{\circ 2}) / (P_{H_2} / P^{\circ})$$

The concentration of the bulk hydrogen species is provided in relative concentration units of θ_B . In practice, θ_B can be set equal to the ratio of bulk hydrogen to palladium, $X = H/Pd$. The standard states used are $P^{\circ}=1$ bar and $\theta^{\circ}=1$.

Effective H Absorption Thermodynamics

Shomate Equations for Gas Phase Thermodynamics of H₂:

The Shomate equations provide a calculation for the gas phase entropy, and also the change in gas phase enthalpy relative to 298.15 K. The empirical constants which go into the equations are below (obtained from the NIST chemistry webbook), followed by the Shomate equations.

$$A_{sh} = 33.066178$$

$$B_{sh} = -11.363417$$

$$C_{sh} = 11.432816$$

$$D_{sh} = -2.772874$$

$$E_{sh} = -0.158558$$

$$F_{sh} = -9.980797$$

$$G_{sh} = 172.707974$$

$$H_{sh} = 0$$

$$T_{kK} = T_K/1000 \quad (\text{temperature in kilo-Kelvin is temperature in Kelvin divided by 1000}).$$

$$DH_{sh298.15}^{\circ} = H^{\circ} - H_{298.15}^{\circ} = A_{sh} * T_{kK} + B_{sh} * T_{kK}^2 / 2 + C_{sh} * T_{kK}^3 / 3 + D_{sh} * T_{kK}^4 / 4 - E_{sh} / T_{kK} + F - H$$

$$S_{sh}^{\circ} = A_{sh} * \ln(T_{kK}) + B_{sh} * T_{kK} + C_{sh} * T_{kK}^2 / 2 + D_{sh} * T_{kK}^3 / 3 - E_{sh} / (2 * T_{kK}^2) + G_{sh}$$

b PdH with large-size energetics

Reference¹⁶ consolidated the data from a number of experimental studies, and good agreement was found between the various studies for the Pd-H phase diagram. The data was fitted to produce effective enthalpy and entropy equations, given standard states of P°=1 bar and θ°= 1. The configurational entropy is included implicitly.

$$DH_{eff} = X * 78880 \text{ J/mol} - 82230 \text{ J/mol}$$

$$DS_{eff} = -139.1 * X \text{ J / (mol K)} + 11.8 \text{ J / (mol K)}$$

Parameterized Equations for direct H/Pd calculation:

$$M_{P\theta} = 0.000234 * T^2 - 0.216 * T + 64.8$$

$$B_{P\theta} = -0.000234 * T^2 + 0.221 * T - 55.1$$

$$\text{Log}(P_{H_2} / 1 \text{ Pa}) = M_{P\theta} * \theta_B + B_{P\theta}$$

b PdH with finite-size energetics

The values obtained for 4 nm particles in reference²⁹ were used as a starting point to obtain equations that brought the 300 K isotherm in agreement with the data collected inside reference.²⁴

$$DH_{eff} = X * 78880 \text{ J/mol} - 91000 \text{ J/mol}$$

$$DS_{eff} = -165 * X \text{ J / (mol K)} - 100 \text{ J / (mol K)}$$

The following variation (“Variation 2”) was also tested, which has a smaller (and constant) change in enthalpy as well as a smaller change in entropy.

$$DH_{\text{eff}} = -18000 \text{ J/mol}$$

$$DS_{\text{eff}} = -165.0 * X * J / (\text{mol K}) - 11.0 J / (\text{mol K})$$

a PdH with large-size energetics

$$DH_{\text{eff}} = 2 * (-10613 + (-80200 * X^4 + 175000 * X^3 - 58000 * X^2 - 28800 * X))$$

$$DS_{\text{eff}} = -S_{\text{sh}}^{\circ}$$

a PdH with finite-size energetics

For the first variation of finite-size energetics effective thermodynamics, the subsurface concentration region where a PdH might exist was not reached. For the second variation of finite-size energetics effective thermodynamics, a single phase treatment was used and the same equations (below) were used even in the low hydrogen concentration range.

$$DH_{\text{eff}} = -18000 \text{ J/mol}$$

$$DS_{\text{eff}} = -165.0 * X * J / (\text{mol K}) - 11.0 J / (\text{mol K})$$

Quantifying Surface and Subsurface Sites

The total number of Pd atoms ($N_{\text{total-Pd}}$) per particle and the number of surface Pd atoms ($N_{\text{surf-Pd}}$) per particle were calculated using the following equations, which are based on fitting Table 1 of ref⁵⁶. These equations are accurate relative to the theoretical values (<1 % error), and were used successfully previously in quantitative kinetic modeling.^{57,59}

$$N_{\text{total-Pd}} = 0.4156633496205 * (D_{\text{part}}^3) + 0.03515070748628 * (D_{\text{part}}^2) + 0.2574185727647 * (D_{\text{part}}) + 0.6278774572401$$

$$N_{\text{surf-Pd}} = 0.00005364289590659 * (D_{\text{part}}^3) + 2.494170821774 * (D_{\text{part}}^2) - 4.869087366743 * (D_{\text{part}}) + 3.777603227496$$

Where D_{part} is the diameter of the particle in units of atomic diameter as defined by the material's Metal-Metal bond distance. For Pd, the metal-metal effective atomic diameter is 0.275, so D_{part} is 36.36 for a 10 nm particle, and this also indicates the number of layers in the particle (36 atomic diameter units indicates a particle with 18 layers of Pd to the center of the particle). The particles may of course differ in shape from the thermodynamic ideal, but this will not be the greatest source of error in calculating the surface to subsurface Pd ratio.

V. Kinetic Equations

All rates of change were modeled using ordinary differential equations.

The surface hydrogen rates were described by standard Langmuir type equations and hopping-diffusion equations, with all terms canceling at true equilibrium or steady-state.

$$\begin{aligned} d\theta_{\text{Surf}}/dt = & k_{\text{ads}} * P_{\text{H}_2} * (1-\theta_{\text{Surf}}) * (1-\theta_{\text{Surf}}) - k_{\text{rec}} * \theta_{\text{surf}} * \theta_{\text{surf}} \\ & + k_{\text{Sub1-Surf}} * \theta_{\text{Sub1}} * (1-\theta_{\text{Surf}}) - k_{\text{Surf-Sub1}} * \theta_{\text{Surf}} * (1-\theta_{\text{Sub1}}) \end{aligned}$$

For each layer of subsurface hydrogen, the rate of change had terms for diffusion towards or away from the bulk, with all terms canceling at true equilibrium or steady-state.

$$\begin{aligned} d\theta_{\text{Sub}_n}/dt = & k_{\text{Sub}_{n-1}\text{-Sub}_n} * \theta_{\text{Sub}_{n-1}} * (1-\theta_{\text{Sub}_n}) - k_{\text{Sub}_n\text{-Sub}_{n-1}} * \theta_{\text{Sub}_n} * (1-\theta_{\text{Sub}_{n-1}}) \\ & + k_{\text{Sub}_{n-2}\text{-Sub}_n} * \theta_{\text{Sub}_{n-2}} * (1-\theta_{\text{Sub}_n}) - k_{\text{Sub}_n\text{-Sub}_{n-2}} * \theta_{\text{Sub}_n} * (1-\theta_{\text{Sub}_{n-2}}) \end{aligned}$$

Where “n” is the layer of subsurface hydrogen (e.g., first layer, second layer, etc.), and all terms are described by coverage dependent Arrhenius form rate constants with the terms from the tables of Energetic and Kinetic Parameters provided.

VI. Surface and Subsurface Hydrogen Populations Simulation Details

Simulations were performed in Athena Visual Studio. The surface to subsurface hydrogen ratios were calculated using ordinary differential equation-based steady-state kinetic simulations. There are three benefits to using this approach: a) the effects of diffusion between different states are included without additional effort because migration between states is modeled explicitly, b) a closed form solution to the ratios of the species does not need to be derived -- the equations which modify the state energies (and transition state energies) are solved numerically and dynamically until a steady-state (also known as a stationary state) is achieved, c) if a steady-state (or stationary state) is not achieved during experimental timescales, that will be readily apparent within the simulations, which will provide both the steady-state resulting ratios along with the resulting ratios after a specified amount of experimental time.

Athena Visual Studio solves ordinary differential equations using an implicit stiff ordinary differential equation solver based on Gear's method.⁵⁸ The software includes the Runge-Kutta method as an option (fourth order), and the results are essentially indistinguishable between the two integration methods. The concentrations of the first subsurface hydrogen layer and the bulk hydrogen (8th subsurface layer and deeper) were described by the thermodynamic or θ -P-T equations. The thermodynamics of the first subsurface hydrogen layer was described in simulations by a sorption enthalpy 6.75 kJ/mol more exothermic (per H atom) than for the bulk hydrogen layer. The concentrations of the surface hydrogen and the bulk-like hydrogen (subsurface hydrogen in layers 2-7) were described using ordinary differential equations and the kinetic parameters from the tables above. Implicit equations for the concentration, of the form $P(\theta)$ rather than $\theta(P)$ were solved using Newton's method. Newton's method was applied iteratively since the equations were coupled. The initial conditions for surface hydrogen and bulk like hydrogen were $\theta=0$. The simulations were constrained to have final concentrations >0 , to prevent negative Surface H to Subsurface H ratios from numerical errors.

The simulations were performed for H₂ gas phase exposure times of 6 minutes, 1 hour, and 10 hours. The rate constants were dynamic during simulation due to the coverage dependences of the terms

within them. Whether the surface hydrogen species reached in equilibrium state (or not) during the simulation time was assessed by taking the rate constants present at the end of simulation for adsorption and desorption and calculating an instantaneous projected equilibrium concentration based on this ratio and the gas phase pressure. If the concentration in the end of simulation was within 5% of the instantaneous projected equilibrium, the surface species in the simulation were considered to have reached equilibrium. Most conditions achieved equilibrium for the surface species within all exposure times (see Figures below): only some low temperature and low pressure conditions did not achieve surface hydrogen equilibrium during these exposure timescales. For the conditions where equilibrium was not achieved for the surface species, the points were plotted in red, though the trends observed suggest that most of the concentrations would be quite similar even if equilibrium was achieved. For these non-equilibrium cases, the differences observed between the different exposure times were small (relative to breadth of the range explored).

VII. Surface and Subsurface Hydrogen Populations Figures and Interpretation

Under the conditions of the experiments conducted here, the bulk species are expected to be b PdH. The surface to subsurface ratio is plotted in Figures S2 to S4 for cases where bulk species are constrained to being b PdH (including when in the coexistence region of the phase diagram). For Figures S2 to S4, the points are in increasing pressure from left to right, and are in increasing temperature from bottom to top (the lowest points are at 15 K, the upper most points are at 215 K, with increments of 20 K between temperatures). Black points represent cases where equilibrium was achieved, and red points represent cases where equilibrium was not achieved for the surface species within one hour of simulation time. The left-hand panels were from simulations using large-size energetics while the right hand panels were from simulations using “finite-size energetics”. Several features can be noticed inside these figures. The finite-size energetics case has higher surface to subsurface hydrogen ratios: this is because the finite-size requires more pressure to achieve the same concentration of subsurface hydrogen. In both types of lattices, higher temperatures achieve a greater surface to subsurface hydrogen ratio when the bulk is constrained to being b PdH. This is because within the model the surface hydrogen state is more stable than the subsurface hydrogen state, in agreement with experiment,²⁸ which means that higher temperatures will deplete the subsurface states more easily than the surface states. Or, in another way of stating it, lower temperatures are required to populate the subsurface hydrogen states, which is an agreement with experiment. It is worth noting that from the trends visible in the figures, it is clear that the cases which did not reach equilibrium within one hour would not shift very much if equilibrium was achieved. As mentioned, simulations were performed for 6 minutes, 1 hour, and 10 hours of exposure time. The 10 hour cases and 1 hour exposure simulations did not show large differences – the inability to reach equilibrium in the red points on this time scale is due to both the very low pressures towards the left of the graphs, and also due to the low temperature’s effect on Arrhenius rate constants.

In Fig. AS5, we explore what happens when the bulk species are considered to convert completely to a PdH inside the coexistence region. The data is plotted on a log log plot. This is because at low pressures the bulk becomes essentially completely depleted of subsurface hydrogen.

The lower right hand portion of the figure is the same as previously for large-size energetics because in that region only the β PdH phase is present. In Fig. AS5, the trend of increasing Surface H / Subsurface H with increasing temperature is still qualitatively true, but not strictly true. For $1\text{E-}9$ bar, the highest point (maximum ratio) is achieved at 160K, for $1\text{E-}7$ bar the maximum ratio is at 240K, for $1\text{E-}3$ bar the maximum ratio is at 300K. These results are in qualitative agreement with experimental data.⁶⁰ It is interesting that these maxima at low pressure are in the same temperature range where subsurface hydrogen effects have been studied for HD exchange and alkene hydrogenation.^{33,61-66} As the energetics varies with particle size and other factors, the results in Fig. AS5 should be regarded as a guideline but not a strict limit when studying Pd particles. Fig. AS6 shows the results using a second variation for the thermodynamics of finite size effects. Experimental data is not currently available to know which is more accurate between Fig. AS6 and the right-hand Panel of Fig. AS3. However, both Fig. AS6 and the right-hand Panel of Fig. AS3 are similar in the atmospheric pressure range, which is the range where the experiments of the current study were conducted.

Fig. AS2: the surface H to subsurface H ratio for 5 nm Pd nanoparticles when the bulk species are constrained to be PdH. Left panel: output using large-size energetics. Right panel: output using “finite-size energetics”. Black points represent cases where equilibrium was achieved, and red points represent cases where equilibrium was not achieved for the surface species within one hour. The points are in increasing pressure from left to right, and are in increasing temperature from bottom to top (the lowest points are at 15K, the upper most points are at 215 K, with increments of 20 K between temperatures).

Fig. AS3: the surface H to subsurface H ratio for 10 nm Pd nanoparticles when the bulk species are constrained to b PdH. Left panel: output using large-size energetics. Right panel: output using “finite-size energetics”. Points as in Fig. AS2. The points are in increasing pressure from left to right, and are in increasing temperature from bottom to top (the lowest points are at 15K, the upper most points are at 215 K, with increments of 20 K between temperatures).

Fig. AS4: the surface H to subsurface H ratio for 15 nm Pd nanoparticles when the bulk species are constrained to b PdH. Left panel: output using large-size energetics. Right panel: output using “finite-size energetics”. Points as in Fig. AS2. The points are in increasing pressure from left to right, and are in increasing temperature from bottom to top (the lowest points are at 15K, the upper most points are at 215 K, with increments of 20 K between temperatures).

Fig. AS5: the surface H to subsurface H ratio for Pd nanoparticles when the bulk species are converted to a PdH in the coexistence region of the phase diagram, and the bulk species are kept as PdH for the region of the phase diagram which is known to be b PdH. Points as in Fig. AS2. Left Panel: using large-size energetics for a 10 nm Pd Particle. Right Panel: Using large-size energetics for a 15 nm Pd nanoparticle. The points are in increasing pressure from left to right, and are in increasing temperature from bottom to top (the lowest points are at 100K, the upper most points are at 300 K, with increments of 20 K between temperatures).

Fig. AS6: the surface H to subsurface H ratio for 10 nm Pd nanoparticles using the second variation of thermodynamics for finite-size effects and the bulk species are described as being a single phase.

Points as in Fig. AS2. The points are in increasing pressure from left to right, and are in increasing temperature from bottom to top (the lowest points are at 100K, the upper most points are at 300 K, with increments of 20 K between temperatures).

VIII. Derivation & Calculation of Stress at Core-Shell Interface if PdH is Formed

VIII.1 Derivation

The diameter of the core is:

$$d_i = \sqrt[3]{\frac{6V_i}{\pi}} \quad (\text{Eq. S1})$$

Where V_i and d_i are the initial volume of the core. When the core expands for ΔV_i the final diameter, d_f of the core will be:

$$d_f = \sqrt[3]{\frac{6V_i}{\pi}} \times \sqrt[3]{\left(1 + \frac{\Delta V_i}{V_i}\right)} \quad (\text{Eq. S2})$$

To simplify the above correlations, we approximate the above correlation using a Taylor expansion in the limit of small volume expansion (near the zero asymptote):

$$d_f \cong \sqrt[3]{\frac{6V_i}{\pi}} \times \left(1 + \frac{\Delta V_i}{3V_i}\right) \quad (\text{Eq. S3})$$

Where $\frac{\Delta V_i}{V_i}$ is the volume expansion of the core. The diameter change of the core in the presence of no external shell can be defined as:

$$d_f - d_i \cong 0.33 \left(\frac{\Delta V_i}{V_i}\right) \sqrt[3]{\frac{6V_i}{\pi}} \quad (\text{Eq. S4a})$$

$$(d_f - d_i)/d_i \cong 0.33 \left(\frac{\Delta V_i}{V_i}\right) \quad (\text{Eq. S4b})$$

The circumferential or hoop stress in the shell can be defined as:

$$\sigma_H = \frac{pd}{2t} \quad (\text{Eq. S5})$$

Where p is the internal pressure, d is the internal diameter, and t is the wall thickness.

The change in volume of the shell due to internal stress can be described as

$$\frac{\Delta V_i}{V_i} = \frac{3\sigma_{RR}d}{4tE_{shell}} \quad (\text{Eq. S6a})$$

Where E_{shell} is the young's modulus of the shell. The radial Strain in the spherical coordinate for a solid sphere can be defined as:

$$\varepsilon\varepsilon_{RR} = \frac{du}{dR} \quad (\text{Eq. S6b})$$

where u is the displacement vector. The stress-strain relations for the solid sphere core can be described as:

$$\sigma_{RR} = \frac{E_{core}}{(1+\nu)(1-2\nu)} \left((1-\nu)\varepsilon\varepsilon_{RR} \right) - \left(\frac{\Delta R_{act}}{1-\nu} \right) \quad (\text{Eq. S7})$$

Considering the outer shell restricts the expansion of the inner core, the force balance at the interface suggests:

$$\sigma_{RR_{core}} = \sigma_{RR_{shell}} \quad (\text{Eq. S8})$$

By substituting the equation 7, 6, 4 into equation 8 the following expression can be obtained:

$$\left[\frac{E_{core}}{(1+\nu)(1-2\nu)} \left(0.33 * 0.5 * (1-\nu) \right) - \left(\frac{0.165d_i}{1-\nu} \right) \right]_{core} = \left[\frac{4tE_{shell}}{3d} \right]_{shell} \quad (\text{Eq. S9})$$

Therefore, the equilibrium interface diameter after the core exert the stress to the shell can be calculated as:

$$d_i = \frac{-\frac{E_{core}}{(1+\nu)(1-2\nu)}(0.165(1-\nu)) \pm \sqrt{\left(\left(\frac{E_{core}}{(1+\nu)(1-2\nu)}(0.165(1-\nu))\right)^2 + 0.66 \left[\frac{4tE_{shell}}{3}\right]_{shell}\right)}{-\left(\frac{0.33}{1-\nu}\right)} \quad (\text{Eq. S10})$$

By substituting the above equilibrium interface diameter equation in equation 6, the stress in the shell can be calculated. When the stress exceeds the fractural limit of the shell material, the shell will crack.

VIII.2 Relevant Literature Values

The following values were obtained from the literature.⁶⁷ We see that the tensile strengths of the iron oxides are typically 10 to 20 MPa, and measured values have not been reported above 40 MPa,⁷⁸ so we take the range of 10 to 40 MPa as the limits. The values reported for the Young's modulus for the iron oxides are generally between 130 and 300 GPa (with only one source reporting the possibility of significantly higher values⁶⁹), and these limits will thus be used during the calculations. The value used for Pd is in line with other literature values, which typically range between 120 and 130 GPa (this range is not enough to affect the conclusions of the present study).⁷⁰

Material	Young's Modulus (GPa)	Tensile Strength (MPa)	Ref.
Pd	128	N/A	70
b PdH	112	N/A	70
FeO	--	≥ 25	78
FeO[110]	126	17.2	67
FeO [001]	197	14.0	67

Fe ₂ O ₃	--	≥ 20	78
Fe ₂ O ₃	230-350	--	69
Fe ₂ O ₃ [01̄12]	194	18.0	67
Fe ₂ O ₃ [0001]	246	19.3	67
Fe ₃ O ₄	150-350	--	69
Fe ₃ O ₄	--	> 20	78
Fe ₃ O ₄ [111]	205	15.0	67
Fe ₃ O ₄ [110]	153	13.2	67
Fe ₃ O ₄ [001]	133	12.5	67

VIII.3 Calculation of Strain at Interface & Possibility of Breaking

The below tables provide the Stress/Strength ratios for the various limits of the Shell's Young's Modulus and Tensile Strength (one table for Pd, and one table for PdH). The values reported below assume a volume expansion of 14% for the core,²³ but the stress strength ratios remained >10 even for core volume expansions of only 1% . In all cases, the Stress/Strength ratio is >>1, indicating that the shell would crack if the core is allowed to expand. When hydrogen is added to Pd to make b PdH, the material's volume and also Young's modulus undergo gradual transitions between the Pd values and the b PdH values.⁷⁰ Thus, during the expansion, the stress would be in between those described in the 2 tables (which would result in cracking of the shell). In considering the possibility of b PdH remaining constricted in the shell, the stress at the interface is the same: whether the strain is the same whether due to Pd(H) expanding or Pd(H) constrained to a compressed volume. Thus, for Pd to absorb hydrogen and convert to b PdH requires cracking the shell. However, as shown in the next section, the thermodynamics of Pd conversion to b PdH is not sufficient to crack the shell, and thus Pd is unable to convert to b PdH.

Using values for Pd for the Core and the various limits of Iron Oxide for the Shell:

Core's Young's Modulus (GPa) [†]	Shell's Young's Modulus (GPa)	Shell Tensile Strength (MPa)	Stress at Interface (MPa)	Stress / Strength
128	130	10	371058	~37000
128	130	40	371058	~9000
128	300	10	9957	~1000
128	300	40	9957	~250

Using values for b PdH for the Core and the various limits of Iron Oxide for the Shell:

Core's Young's Modulus (GPa)	Shell's Young's Modulus (GPa)	Shell Tensile Strength (MPa)	Stress at Interface (MPa)	Stress / Strength
112	130	10	36075	~3600
112	130	40	36075	~900
112	300	10	7971	~800
112	300	40	7971	~200

IX. Calculation of Energetics for Core Conversion to PdH vs. Energy Needed to Crack the Iron Oxide Shell

To assess the thermodynamic feasibility of PdH formation and the shell breaking, we make estimates of the amount of energy that would be released by PdH formation, versus what is necessary to crack the shell. As described in the quantification of sites in section IV of this supporting information, we are able to estimate the number of surface sites and subsurface sites for hydrogen adsorption/absorption. For a 5nm Pd particle, the number of surface sites is ~740 and the number of bulk sites is ~1775. The surface hydrogens energy of adsorption is approximately ~40 kJ/mol, while the subsurface hydrogen energy of absorption is ~20 kJ/mol when saturated. Thus, the maximum energy is ~ 35 MJ/mol for the subsurface hydrogens, where the mol now refers to moles of core-shells particles (even if the surface hydrogen's adsorption energy could contribute to the shell's cracking, the total energy for hydride formation would still be < ~ 65 MJ/mol). What about calculating the energy required to crack the shell? The energy for dissociation of each FeO bond is ~409 kJ/mol.⁷¹ We can calculate the number of FeO bonds which need to be broken in order to crack the shell: the circumference of a of a 5nm particle is 15.71 nm, and each Fe-O bond is ~ 2 Angstroms.⁷⁹ Thus, the circumference is ~ 80 Fe-O bond-lengths. The shell is ~2-3 nm thick, which corresponds to 5+ layers of iron oxide.⁶⁷ Our order of magnitude estimate now becomes: $400 \times 80 \times 5 = 160,000 \text{ kJ/mol} = 160 \text{ MJ/mol}$ for the energy to crack the shell, where mol refers to moles of core-shell particles.

References:

- 1 Matsubu, J. C. *et al.* Adsorbate-mediated strong metal-support interactions in oxide-supported Rh catalysts. *Nat. Chem.* **9**, 120-127, doi:10.1038/nchem.2607 (2017).
- 2 Tang, H. *et al.* Strong Metal-Support Interactions between Gold Nanoparticles and Nonoxides. *J. Am. Chem. Soc.* **138**, 56-59, doi:10.1021/jacs.5b11306 (2016).

- 3 Macino, M. *et al.* Tuning of catalytic sites in Pt/TiO₂ catalysts for the chemoselective hydrogenation of 3-nitrostyrene. *Nat. Catal.*, doi:10.1038/s41929-019-0334-3 (2019).
- 4 Wu, C. H. *et al.* Bimetallic synergy in cobalt–palladium nanocatalysts for CO oxidation. *Nat. Catal.* **2**, 78-85, doi:10.1038/s41929-018-0190-6 (2018).
- 5 Guo, Z., Kang, X., Zheng, X., Huang, J. & Chen, S. PdCu alloy nanoparticles supported on CeO₂ nanorods: Enhanced electrocatalytic activity by synergy of compressive strain, PdO and oxygen vacancy. *J. Catal.* **374**, 101-109, doi:10.1016/j.jcat.2019.04.027 (2019).
- 6 Kast, P. *et al.* CO oxidation as a test reaction for strong metal–support interaction in nanostructured Pd/FeO powder catalysts. *Appl. Catal. A Gen.* **502**, 8-17, doi:10.1016/j.apcata.2015.04.010 (2015).
- 7 Wu, C.-T. *et al.* A non-syn-gas catalytic route to methanol production. *Nat. Commun.* **3**, doi:10.1038/ncomms2053 (2012).
- 8 Benziger, J. B. & Larson, L. R. An infrared spectroscopy study of the adsorption of CO on Fe/MgO. *J. Catal.* **77**, 550-553, doi:10.1016/0021-9517(82)90195-6 (1982).
- 9 Felicissimo, M. P., Martyanov, O. N., Risse, T. & Freund, H.-J. Characterization of a Pd–Fe bimetallic model catalyst. *Surf. Sci.* **601**, 2105-2116, doi:10.1016/j.susc.2007.02.023 (2007).
- 10 Wei, X., Ma, Z., Lu, J., Mu, X. & Hu, B. Strong metal–support interactions between palladium nanoclusters and hematite toward enhanced acetylene dicarbonylation at low temperature. *New J. Chem.* **44**, 1221-1227, doi:10.1039/C9NJ05493F (2020).
- 11 Liu, F. *et al.* Exchange-coupled fct-FePd/alpha-Fe nanocomposite magnets converted from Pd/Fe₃O₄ core/shell nanoparticles. *Chemistry* **20**, 15197-15202, doi:10.1002/chem.201403787 (2014).
- 12 Xu, W. *et al.* Nanoporous Palladium Hydride for Electrocatalytic N₂ Reduction under Ambient Conditions. *Angew. Chem. Int. Ed.* **59**, 3511-3516, doi:10.1002/anie.201914335 (2020).
- 13 Bardhan, R. *et al.* Uncovering the intrinsic size dependence of hydriding phase transformations in nanocrystals. *Nat. Mater.* **12**, 905-912 (2013).
- 14 Mobus, K., Grunewald, E., Wieland, S. D., Parker, S. F. & Albers, P. W. Palladium-catalyzed selective hydrogenation of nitroarenes: Influence of platinum and iron on activity, particle morphology

and formation of beta-palladium hydride. *J. Catal.* **311**, 153-160, doi:10.1016/j.jcat.2013.11.019 (2014).

15 Lewis, F. A. The palladium-hydrogen system: Structures near phase transition and critical points. *Int. J. Hydrogen Energ.* **20**, 587-592, doi:10.1016/0360-3199(94)00113-E (1995).

16 Manchester, F., San-Martin, A. & Pitre, J. The H-Pd (hydrogen-palladium) system. *J. Phase Equilib.* **15**, 62-83 (1994).

17 Flanagan, T. B. & Oates, W. A. The Palladium-Hydrogen System. *Annu. Rev. Mater. Sci.* **21**, 269-304 (1991).

18 Lacher, J. R. A Theoretical Formula for the Solubility of Hydrogen in Palladium. *P. Roy. Soc. A-Math. Phy.* **161**, 525-545 (1937).

19 Wadell, C. *et al.* Thermodynamics of hydride formation and decomposition in supported sub-10 nm Pd nanoparticles of different sizes. *Chem. Phys. Lett.* **603**, 75-81, doi:10.1016/j.cplett.2014.04.036 (2014).

20 Frieske, H. & Wicke, E. Magnetic Susceptibility and Equilibrium Diagram of PdH_n. *Berichte der Bunsengesellschaft für physikalische Chemie* **77**, 48-52, doi:10.1002/bbpc.19730770112 (1973).

21 Lynch, J. F. & Flanagan, T. B. Investigation of Dynamic Equilibrium between Chemisorbed and Absorbed Hydrogen in Palladium-Hydrogen System. *J. Phy. Chem.* **77**, 2628-2634 (1973).

22 Flanagan, T. B. & Wang, D. Hydrogen Permeation through fcc Pd–Au Alloy Membranes. *The J. Phy. Chem. C* **115**, 11618-11623, doi:10.1021/jp201988u (2011).

23 Marcano Romero, D. A. *Computer Simulations for Hydrogen Loaded Palladium Clusters*, Georg-August University (2007).

24 Shtaya-Suleiman, M. A. *Size-selective synthesis of nanometersized Palladium clusters and their hydrogen solvation behaviour*, Ph. D Thesis, Göttingen University, Göttingen, Switzerland, (2003).

25 Kirchheim, R. Solid solutions of hydrogen in complex materials. *Solid State Physics-Advances in Research and Applications* **59**, 203-292 (2004).

26 Baldi, A., Narayan, T. C., Koh, A. L. & Dionne, J. A. In situ detection of hydrogen-induced phase transitions in individual palladium nanocrystals. *Nat Mater* **13**, 1143-1148, doi:10.1038/nmat4086

(2014).

27 Pundt, A. *et al.* Hydrogen and Pd-clusters. *Mater. Sci. Eng. B* **108**, 19-23, doi:10.1016/j.mseb.2003.10.029 (2004).

28 Wilde, M., Fukutani, K., Naschitzki, M. & Freund, H. J. Hydrogen absorption in oxide-supported palladium nanocrystals. *Phy. Rev. B* **77**, - (2008).

29 Crespo, E., Claramonte, S., Ruda, M. & de Debiaggi, S. R. Thermodynamics of hydrogen in Pd nanoparticles. *Int.J Hhydrogen Energ.* **35**, 6037-6041 (2010).

30 Wilde, M. & Fukutani, K. Penetration mechanisms of surface-adsorbed hydrogen atoms into bulk metals: Experiment and model. *Phy. Rev. B* **78**, - (2008).

31 Hong, S. & Rahman, T. S. Adsorption and diffusion of hydrogen on Pd(211) and Pd(111): Results from first-principles electronic structure calculations. *Phy. Rev. B* **75**, - (2007).

32 Ward, T. L. & Dao, T. Model of hydrogen permeation behavior in palladium membranes. *J. Membr. Sci.* **153**, 211-231, doi:10.1016/S0376-7388(98)00256-7 (1999).

33 Savara, A., Ludwig, W. & Schauermaun, S. Kinetic Evidence for a Non-Langmuir-Hinshelwood Surface Reaction: H/D Exchange over Pd Nanoparticles and Pd(111). *Chemphyschem* **14**, 1686-1695 (2013).

34 Johansson, M. *et al.* Hydrogen adsorption on palladium and palladium hydride at 1 bar. *Surf. Sci.* **604**, 718-729 (2010).

35 Nobuhara, K., Kasai, H., Dino, W. A. & Nakanishi, H. H-2 dissociative adsorption on Mg, Ti, Ni, Pd and La surfaces. *Surf. Sci.* **566**, 703-707 (2004).

36 Comsa, G., David, R. & Schumacher, B.-J. Fast deuterium molecules desorbing from metals. *Surf. Sci.* **95**, L210-L216, doi:10.1016/0039-6028(80)90121-1 (1980).

37 Padama, A. A. B., Chantaramolee, B., Nakanishi, H. & Kasai, H. Hydrogen atom absorption in hydrogen-covered Pd(110) (1 × 2) missing-row surface. *Int. J. Hydrogen Energ.* **39**, 6598-6603, doi:10.1016/j.ijhydene.2014.02.019 (2014).

38 Deveau, N. D., Ma, Y. H. & Datta, R. Beyond Sieverts' law: A comprehensive microkinetic model of hydrogen permeation in dense metal membranes. *J. Membrane Sci.* **437**, 298-311 (2013).

- 39 Paul, J. & Sautet, P. Density-functional periodic study of the adsorption of hydrogen on a palladium (111) surface. *Phys Rev B Condens Matter* **53**, 8015-8027, doi:10.1103/physrevb.53.8015 (1996).
- 40 Gladys, M. J. *et al.* Comparison of hydrogen and deuterium adsorption on Pd(100). *J. Chem. Phys.* **132**, 024714, doi:10.1063/1.3292686 (2010).
- 41 Ferrin, P., Kandoi, S., Nilekar, A. U. & Mavrikakis, M. Hydrogen adsorption, absorption and diffusion on and in transition metal surfaces: A DFT study. *Surf. Sci.* **606**, 679-689, doi:10.1016/j.susc.2011.12.017 (2012).
- 42 Gross, A. Ab initio molecular dynamics simulations of the adsorption of H₂ on palladium surfaces. *ChemPhysChem* **11**, 1374-1381, doi:10.1002/cphc.200900818 (2010).
- 43 Conrad, H., Ertl, G. & Latta, E. E. Adsorption of Hydrogen on Palladium Single-Crystal Surfaces. *Surf. Sci.* **41**, 435-446 (1974).
- 44 Kozlov, S. M., Aleksandrov, H. A. & Neyman, K. M. Energetic Stability of Absorbed H in Pd and Pt Nanoparticles in a More Realistic Environment. *J. Phy. Chem. C* **119**, 5180-5186, doi:10.1021/jp513022m (2015).
- 45 Paul, J. F. & Sautet, P. Density-functional periodic study of the adsorption of hydrogen on a palladium (111) surface. *Phy. Rev. B* **53**, 8015-8027 (1996).
- 46 Nobuhara, K., Kasai, H., Nakanishi, H. & Okiji, A. Coverage dependence of hydrogen absorption into Pd(111). *J. Appl. Phys.* **92**, 5704-5706 (2002).
- 47 Rick, S. W., Lynch, D. L. & Doll, J. D. The quantum dynamics of hydrogen and deuterium on the Pd(111) surface: A path integral transition state theory study. *J. Chem. Phys.* **99**, 8183-8193, doi:10.1063/1.465645 (1993).
- 48 Greeley, J. & Mavrikakis, M. Surface and subsurface hydrogen: Adsorption properties on transition metals and near-surface alloys. *J. Phys. Chem. B* **109**, 3460-3471 (2005).
- 49 Kurokawa, H. *et al.* Monte Carlo simulation of hydrogen absorption in palladium and palladium-silver alloys. *Catal. Today* **82**, 233-240, doi:http://dx.doi.org/10.1016/S0920-5861(03)00237-2 (2003).
- 50 Ke, X. Z. & Kramer, G. J. Absorption and diffusion of hydrogen in palladium-silver alloys by

density functional theory. *Phys. Rev. B* **66**, - (2002).

51 Libowitz, G. G. in *Nonstoichiometric Compounds* Vol. 39 *Advances in Chemistry* Ch. 7, 74-86 (AMERICAN CHEMICAL SOCIETY, 1963).

52 Farkas, A. On the rate determining step in the diffusion of hydrogen through palladium. *Transactions of the Faraday Society* **32**, 1667-1679, doi:10.1039/TF9363201667 (1936).

53 Kay, B. D., Peden, C. H. F. & Goodman, D. W. Kinetics of hydrogen absorption by Pd(110). *Physical Review B* **34**, 817-822 (1986).

54 Salloum, M., James, S. C. & Robinson, D. B. Effects of surface thermodynamics on hydrogen isotope exchange kinetics in palladium: Particle and flow models. *Chem. Eng. Sci.* **122**, 474-490, doi:10.1016/j.ces.2014.09.001 (2015).

55 Morreale, B. D. *et al.* The permeability of hydrogen in bulk palladium at elevated temperatures and pressures. *J. Membrane Sci.* **212**, 87-97, doi:10.1016/S0376-7388(02)00456-8 (2003).

56 Poole, C. P. & Owens, F. J. *Introduction to Nanotechnology*. (John Wiley & Sons, 2003).

57 Savara, A., Rossetti, I., Chan-Thaw, C. E., Prati, L. & Villa, A. Microkinetic Modeling of Benzyl Alcohol Oxidation on Carbon-Supported Palladium Nanoparticles. *Chemcatchem* **8**, 2482-2491, doi:10.1002/cctc.201600368 (2016).

58 Savara, A. Simulation and fitting of complex reaction network TPR: The key is the objective function. *Surf. Sci.* **653**, 169-180, doi:10.1016/j.susc.2016.07.001 (2016).

59 Savara, A. *et al.* Molecular Origin of the Selectivity Differences between Palladium and Gold–Palladium in Benzyl Alcohol Oxidation: Different Oxygen Adsorption Properties. *Chemcatchem* **9**, 253-257, doi:10.1002/cctc.201601295 (2017).

60 Gdowski, G. E., Felter, T. E. & Stulen, R. H. Effect of Surface-Temperature on the Sorption of Hydrogen by Pd(111). *Surf. Sci.* **181**, L147-L155, doi:10.1016/0039-6028(87)90187-7 (1987).

61 Schauermaun, S., Ludwig, W., Savara, A. & Freund, H. J. Role of subsurface hydrogen diffusion in hydrocarbon conversions on supported model catalysts: A molecular beam study. *Abstr. Pap. Am. Chem. Soc.* **243** (2012).

62 Savara, A., Ludwig, W., Dostert, K. H. & Schauermaun, S. Temperature dependence of the

- 2-butene hydrogenation over supported Pd nanoparticles and Pd(111). *J. Mol. Catal. A Chem.* **377**, 137-142 (2013).
- 63 Ludwig, W., Savara, A., Schauermaun, S. & Freund, H. J. Role of Low-Coordinated Surface Sites in Olefin Hydrogenation: A Molecular Beam Study on Pd Nanoparticles and Pd(111). *Chemphyschem* **11**, 2319-2322 (2010).
- 64 Ludwig, W., Savara, A. & Schauermaun, S. Role of hydrogen in olefin isomerization and hydrogenation: a molecular beam study on Pd model supported catalysts. *Dalton Trans.* **39**, 8484-8491, doi:10.1039/C003133J (2010).
- 65 Ludwig, W., Savara, A., Madix, R. J., Schauermaun, S. & Freund, H. J. Subsurface Hydrogen Diffusion into Pd Nanoparticles: Role of Low-Coordinated Surface Sites and Facilitation by Carbon. *J. Phys. Chem. C* **116**, 3539-3544 (2012).
- 66 Ludwig, W., Savara, A., Brandt, B. & Schauermaun, S. A kinetic study on the conversion of cis-2-butene with deuterium on a Pd/Fe₃O₄ model catalyst. *Phys. Chem. Chem. Phys.* **13**, 966-977, doi:10.1039/C0CP00078G (2011).
- 67 Liao, P. & Carter, E. A. Ab initio DFT + U predictions of tensile properties of iron oxides. *J. Mater. Chem.* **20**, 6703-6719, doi:10.1039/c0jm01199a (2010).
- 68 Hidaka, Y., Anraku, T. & Otsuka, N. Deformation of Iron Oxides upon Tensile Tests at 600–1250°C. *Oxid. Met.* **59**, 97-113, doi:10.1023/a:1023070016230 (2003).
- 69 Chicot, D. *et al.* Mechanical properties of magnetite (Fe₃O₄), hematite (α-Fe₂O₃) and goethite (α-FeO·OH) by instrumented indentation and molecular dynamics analysis. *Materials Chemistry and Physics* **129**, 862-870, doi:10.1016/j.matchemphys.2011.05.056 (2011).
- 70 Fabre, A., Finot, E., Demoment, J. & Contreras, S. In situ measurement of elastic properties of PdH_x, PdD_x, and PdTx. *J. Alloy. Comp.* **356-357**, 372-376, doi:10.1016/S0925-8388(03)00269-X (2003).
- 71 Speight, J. *Lange's Handbook of Chemistry, Seventeenth Edition.* (McGraw-Hill Education, 2016).
- 72 Güttel, R., Paul, M., Galeano, C. & Schüth, F. Au, @ZrO₂ yolk-shell catalysts for CO oxidation:

Study of particle size effect by ex-post size control of Au cores. *J. Catal.* **289**, 100-104, doi:10.1016/j.jcat.2012.01.021 (2012).

73 Galeano, C. *et al.* Yolk-Shell Gold Nanoparticles as Model Materials for Support-Effect Studies in Heterogeneous Catalysis: Au, @C and Au, @ZrO₂ for CO Oxidation as an Example. *Chem. Eur. J.* **17**, 8434-8439, doi:10.1002/chem.201100318 (2011).

74 Güttel, R., Paul, M. & Schüth, F. Activity improvement of gold yolk-shell catalysts for CO oxidation by doping with TiO₂. *Catal. Sci. Technol.* **1**, 65-68, doi:10.1039/c0cy00026d (2011).

75 Pundt, A. *et al.* Hydrogen and Pd-clusters. *Mater. Sci. Eng. B* **108**, 19-23, doi:10.1016/j.mseb.2003.10.029 (2004).

76 Gladys, M. J. *et al.* Comparison of hydrogen and deuterium adsorption on Pd(100). *The J. Chem. Phys.* **132**, 024714, doi:10.1063/1.3292686 (2010).

77 Groß, A. Ab Initio Molecular Dynamics Simulations of the Adsorption of H₂ on Palladium Surfaces. *ChemPhysChem* **11**, 1374-1381, doi:10.1002/cphc.200900818 (2010).

78 Hidaka, Y., Anraku, T. & Otsuka, N. Deformation of Iron Oxides upon Tensile Tests at 600–1250°C. *Oxid. Met.* **59**, 97-113, doi:10.1023/a:1023070016230 (2003).

79 Levy, D., Giustetto, R. & Hoser, A. Structure of magnetite (Fe₃O₄) above the Curie temperature: a cation ordering study. *Phys. Chem. Miner.* **39**, 169-176, doi:10.1007/s00269-011-0472-x (2012).

REVIEWER COMMENTS

Reviewer #1 (Remarks to the Author):

After the modifications, I think that this work might be accepted.

Reviewer #2 (Remarks to the Author):

The authors have done an extensive effort to address successfully the majority of issues raised by all reviewers. I do have some remaining concerns.

1. I cannot agree with that SMSIR is realized by stating from the final architecture of SMSI (full encapsulated core-shell structure). As the phenomenon in SMSI state is not just encapsulation, in which there is some new bond formed, such as Pt-Ti bond in Pt/TiO₂. Therefore, the core-shell structure is not the final architecture of SMSI.

2. About the latest research progress of SMSI, the authors may also give an attention the article (Nature Communications, 2019, 10 5790), in which the permeable TiO_x thin layer fully encapsulates Au nanoparticles to prevent the dissolution, disintegration, and aggregation of active sites during catalysis.

3. Insufficient evidence for electron transfer, now only the state of palladium was characterized. Regarding the state of iron, such as the Pd-Fe bonding as well as the partial intercalation of palladium into iron oxide, all are only speculated without any direct experimental results. Mössbauer spectroscopy is the most effective method to characterize the chemical state of iron, which can selectively give the iron information in various samples after different temperature reduction and reoxidation.

4. What will happen when Pd-Fe₃O₄-Re is reduced in H₂ at 300 °C for 1 h and whether there is some difference between the structure of resultant and Pd-Fe₃O₄-H.

5. It should be “Fe₃O₄ islands on Pd” not “Pd islands on Fe₃O₄” in Q3.

Reviewer #3 (Remarks to the Author):

Authors have revised the manuscript substantially and have addressed all the comments satisfactorily. Now this article can be accepted for publication in Nature Communications.

Incl: Our answers to reviewer's questions

Reviewer #2

Comments: The authors have done an extensive effort to address successfully the majority of issues raised by all reviewers. I do have some remaining concerns.

Response: We greatly appreciate your comments and have revised the manuscript accordingly. Please see below for details.

Q1: I cannot agree with that SMSIR is realized by stating from the final architecture of SMSI (full encapsulated core-shell structure). As the phenomenon in SMSI state is not just encapsulation, in which there is some new bond formed, such as Pt-Ti bond in Pt/TiO₂. Therefore, the core-shell structure is not the final architecture of SMSI.

Response: Thank you for your question. As you mentioned, with the formation of SMSI, not only encapsulation can be formed, new bonds and charge transfer are always found as well. We are well aware of this concept of SMSI. That is why we performed the detailed experimental and theoretical analysis to demonstrate both the morphological (encapsulation) and the electronic (in our case, new Pd-Fe bond formation and electron transfer between Pd and Fe₃O₄) characters of SMSI in our Pd-Fe₃O₄ system. We emphasize that the SMSI featured by Pd-Fe₃O₄ is related into two main aspects: 1) encapsulation which in our case is the encapsulation of Pd by Fe₃O₄ shell; even though the hydrogen treated one demonstrated an intermediate state, it still presents the feature of SMSI (the coverage of metal by metal oxide). This intermediate state actually benefits the following catalytic study because it can provide much more abundant active sites than conventional SMSI with complete encapsulation; 2) the electron transfer between Pd and Fe₃O₄ and new bond formation of Pd-Fe which was proved by our XAFS, XPS, and CO DRIFTS investigations.

The strategy that we employed here to construct SMSI is starting from the final architecture of SMSI (coverage of metal by metal oxide), not the final state of SMSI. We didn't mean to refer the starting core-shell structure as the final SMSI state. It rather just demonstrates that the encapsulation (core-shell structure) is the final morphology of SMSI, and with the treatment of the core-shell NPs in H₂ to obtain Pd-Fe₃O₄-H, the morphology turns from core-shell to yolk-shell, and numerous voids are detected in the shell. Especially, with the formation of SMSIR, Pd intercalated into the matrix of Fe₃O₄ shell, which was confirmed by XAFS results, indicating the formation of strong interactions between Pd and Fe₃O₄.

Thanks for bringing this up. In order to avoid any confusion or misunderstanding, we have changed the expression of "the final architecture" to "the final morphology" in the newly-revised manuscript. SMSIR is realized by stating from the final morphology of SMSI (full encapsulated core-shell structure), and all corresponding expression in the revised manuscript has been changed to be "morphology".

Q2: About the latest research progress of SMSI, the authors may also give an attention the article (Nature Communications, 2019, 10 5790), in which the permeable TiO_x thin layer

fully encapsulates Au nanoparticles to prevent the dissolution, disintegration, and aggregation of active sites during catalysis.

Response: Thank you for your comment. The article (Nature Communications, 2019, 105790) is a very important work in engineering SMSI over Au-based catalysts. The work has been cited and discussed as follows and some other related creative works have been cited (Refs 24 and 25).

“Consequently, it is extremely challenging for some metals, e.g., Au, to manifest SMSI due to their low work function or surface energy²³⁻²⁴. Efforts have been devoted in hope to expand upon the conventional SMSI. One critical element in this pursuit is switching the high-temperature treatment in H₂ into other conditions and thereby changes the mechanistic pathways for the formation of SMSI. For example, Wang et al. reported a SMSI between Au nanoparticles and TiO₂, which is induced by melamine under oxidative atmosphere. With the formation of SMSI, the Au nanoparticles are encapsulated by a permeable TiO_x thin layer, making the Au nanoparticles ultrastable at 800 °C²⁵.”

Q3: Insufficient evidence for electron transfer, now only the state of palladium was characterized. Regarding the state of iron, such as the Pd-Fe bonding as well as the partial intercalation of palladium into iron oxide, all are only speculated without any direct experimental results. Mössbauer spectroscopy is the most effective method to characterize the chemical state of iron, which can selectively give the iron information in various samples after different temperature reduction and reoxidation.

Response: Thanks for your question. To determine the electron transfer between Pd and Fe₃O₄, XPS analysis was performed and it was found that with the formation of SMSIR, charge transfers from Pd to Fe₃O₄, consistent with our WT EXAFS data of Pd-Fe bond formation (**Figure A1**). In the previous revision, we provided the XPS spectra of Pd, and we found that the Pd is positively charged. To make the charge transfer more clearly, XPS spectra of Fe are provided in the revised manuscript. It can be seen from the result that compared with Pd-Fe₃O₄-A, the Fe 2p peak of Pd-Fe₃O₄-H showed a downshift of ~0.3 eV, indicating that the Fe₃O₄ shell is negatively charged with the formation of SMSIR (**Figure A2**). Combined with the Pd results (**Figure A3**), it can be concluded that the charge transfers from Pd to Fe₃O₄ with the formation of SMSIR. Meanwhile, the CO DRIFTS results provided a consistent conclusion. As shown in **Figure A4**, a peak at ~2153 cm⁻¹ was detected both in the CO DRIFTS of Pd-Fe₃O₄-H and Pd-Fe₃O₄-A, which is assigned to Fe³⁺-CO¹. Due to the core-shell morphology of Pd-Fe₃O₄-A where Pd is fully encapsulated by Fe₃O₄, no obvious peak was detected in the CO DRIFTS of Pd-Fe₃O₄-A. In contrast, a very weak peak at 2102 cm⁻¹ can be seen in the CO DRIFTS of Pd-Fe₃O₄-H, which is assigned to the linear-CO adsorption on metallic Pd². More interestingly, an additional peak at 2134 cm⁻¹ can be found. Compared with the linear-CO adsorption on metallic Pd, this peak blueshifted, being assigned to linear-CO adsorption on positively-charged Pd (CO-Pd^{δ+})³. Combined with the XPS analysis, this peak may be attributed to the linear-CO adsorption on Pd^{δ+} in newly emerged Pd-Fe bond in Pd-Fe₃O₄-H.

In summary, the state of Fe has been characterized using XPS, XAFS, and CO-DRIFTS.

Figure A1 Pd K-edge WT EXAFS of Pd-Fe₃O₄-H

Figure A2 High-resolution Fe 2p XPS spectra of FeO_x NPs, Pd-Fe₃O₄-H and Pd-Fe₃O₄-A

Figure A3 High-resolution Pd 3d XPS spectra of Pd-FeO_x NPs, Pd-Fe₃O₄-H and Pd-Fe₃O₄-A.

Figure A4 CO DRIFTS of Pd-Fe₃O₄-H and Pd-Fe₃O₄-A.

Moreover, partial intercalation of Pd into Fe₃O₄ is not a speculated conclusion. We obtained the conclusion from the XAFS results. DFT simulations combined with the EXAFS curve-fitting were carried out to provide more insight into the crystal structure of iron oxide and the interactions between Pd and Fe₃O₄. First, a series of models including a Pd cluster atop the surfaces of Fe₂O₃ and Fe₃O₄, and a Pd cluster in oxygen vacancy of

Fe₂O₃, and Fe₃O₄ surfaces were constructed and optimized by DFT in **Figure A5**, and the corresponding FEFF calculated scattering paths were also presented (**Tables A1, A2, and A3**). Then, the EXAFS curve fitting on the DFT optimized structures (**Figures A6, A7 and Tables A4, A5**) of both Pd K-edge EXAFS and Fe K-edge EXAFS were obtained. It can be concluded from the results that the best-fitted structure of Pd-Fe₃O₄-H is that the Pd atoms intercalate into the Fe₃O₄ matrix (Fig. 4c, detailed optimizing process see Supplementary Fig. 10), indicating that the Pd enters into the Fe₃O₄ lattice, substituting an oxygen vacancy and tends to form the Fe-Pd bond with Fe in Fe₃O₄. **This observation in addition to the previously mentioned XPS and CO DRIFTS suggests that there exist strong interactions between Pd and Fe₃O₄ in Pd-Fe₃O₄-H with the formation of Pd-Fe bond. In contrast, the Pd-Fe₃O₄-A demonstrated a good match to a local geometry of Pd atoms situated on the surface of Fe₃O₄.**

Figure A5 DFT optimized structures of palladium atoms (blue) with different iron oxide (purple-iron, red-oxygen). Two different scenarios for palladium were applied on Fe₃O₄ and Fe₂O₃. All palladium atoms are on the Fe₃O₄ or Fe₂O₃ (Left top and bottom, respectively) and a palladium atom (green color) is in the Fe₃O₄ or Fe₂O₃ substituted oxygen atom from the iron oxide (Right top and bottom, respectively). Highlighted atoms (green-Pd, yellow-Fe) are used as a core atom for Fe K-edge and Pd K-edge, respectively to generate the scattering path.

Table A1 FEFF calculated scattering path generated by using simulated structure model (Pd on Fe₂O₃)

No.	Pd K-edge		Fe K-edge	
	Scattering Path	R _{eff}	Scattering Path	R _{eff}
1	Pd-O1	2.093	Fe-O1	1.898
2	Pd-O2	2.169	Fe-O2	1.933
3	Pd-Pd1	2.529	Fe-O3	2.010
4	Pd-Pd2	2.603	Fe-O-O	3.479
5	Pd-Fe	2.705	Fe-Fe	3.492

Table A2 FEFF calculated scattering path generated by using simulated structure model (Pd on Fe₃O₄)

No.	Pd K-edge		Fe K-edge	
	Scattering Path	R _{eff}	Scattering Path	R _{eff}
1	Pd-O1	2.039	Pd-O1	1.967
2	Pd-Pd	2.528	Pd-O2	2.012
3	Pd-Fe1	2.862	Pd-O3	2.099
4	Pd-Fe2	2.997	Fe-O4	2.128
5	Pd-O2	3.063	Fe-Fe1	2.919

Table A3 FEFF calculated scattering path in Pd K-edge generated by using simulated structure model

No.	Pd in Fe ₂ O ₃		Pd in Fe ₃ O ₄	
	Scattering Path	R _{eff}	Scattering Path	R _{eff}
1	Pd-O1	2.542	Pd-Fe1	2.500
2	Pd-Pd1	2.623	Pd-Fe2	2.582
3	Pd-Fe	2.671	Pd-Fe3	2.683
4	Pd-Pd2	2.665	Pd-Pd1	2.750
5	Pd-Pd3	2.764	Pd-O	2.950

Figure A6 Pd K-edge EXAFS fitting of Pd-Fe₃O₄-H. a) Pd on Fe₂O₃ surface; b) Pd on Fe₃O₄ surface; and c) Pd in Fe₃O₄ surface.

Figure A7 Pd K-edge EXAFS fitting of Pd-Fe₃O₄-A. a) Pd in Fe₂O₃ surface; b) Pd on Fe₂O₃ surface; and c) Pd on Fe₃O₄ surface.

Table A4 Curve-fitting results of Pd K-edge EXAFS spectra for Pd-Fe₃O₄-H using simulated DFT structure model (ΔR = effective bond distance difference, σ^2 = mean-squared relative displacement, parentheses = error)

Simulated Model	Pd on Fe ₂ O ₃	Pd on Fe ₃ O ₄	Pd in Fe ₃ O ₄
E ₀	3.9 (0.22)	4.01 (0.33)	4.13 (0.24)
S ₀ ²	1.24	1.98	1.54
R-factor	0.042	0.019	0.025
ΔR_1	-0.001(0.005)	0.003(0.003)	0.006(0.003)
σ_1^2	0.002(0.0002)	0.001(0.0005)	0.004(0.005)
ΔR_2	0.003(0.012)	-0.004(0.002)	-0.003(0.005)
σ_2^2	0.010(0.018)	0.009(0.002)	0.006(0.014)
ΔR_3	-0.002(0.005)	-0.001(0.042)	0.005(0.081)
σ_3^2	0.082(0.007)	0.003(0.008)	0.001(0.002)
ΔR_4	0.002(0.003)	0.042(0.087)	0.003(0.002)
σ_4^2	0.006(0.002)	0.003(0.001)	0.009(0.005)
ΔR_5	-0.02(0.008)	0.012(0.005)	-0.001(0.008)
σ_5^2	0.003(0.009)	0.001(0.008)	0.012(0.001)

Table A5 Curve-fitting results of Pd K-edge EXAFS spectra for Pd-Fe₃O₄-A using simulated DFT structure model (ΔR = effective bond distance difference, σ^2 = mean-squared relative displacement, parentheses = error)

Simulated Model	Pd on Fe ₂ O ₃	Pd on Fe ₃ O ₄	Pd in Fe ₃ O ₄
E ₀	5.92 (0.18)	4.73 (0.76)	9.35 (0.31)
S ₀ ²	1.32	1.09	0.79
R-factor	0.029	0.015	0.031
ΔR_1	0.001(0.002)	0.003(0.001)	0.033(0.042)
σ_1^2	0.007(0.013)	0.003(0.0001)	0.016(0.002)
ΔR_2	-0.042(0.013)	-0.002(0.001)	-0.015(0.035)
σ_2^2	0.001(0.039)	0.002(0.001)	0.097(0.018)
ΔR_3	0.007(0.001)	-0.001(0.042)	0.029(0.062)
σ_3^2	0.006(0.002)	0.003(0.003)	0.002(0.008)
ΔR_4	0.006(0.001)	0.001(0.007)	0.007(0.001)
σ_4^2	0.003(0.001)	0.007(0.002)	0.003(0.001)
ΔR_5	-0.020(0.005)	0.007(0.007)	0.02(0.008)
σ_5^2	0.002(0.001)	0.003(0.001)	0.012(0.001)

* Mixed model used first 4 scattering paths from Pd on Fe₂O₃ and Pd on Fe₃O₄

We understand that Mössbauer spectroscopy is a powerful tool to determine the chemical state of iron. However, our lab does not have the capability of the Mössbauer spectroscopy, and all labs of our collaborators are closed because of the serious COVID-19 around the world. Since we have well clarified the structure of our prepared samples by a series of characterizations including STEM, XRD, XAFS, XPS, CO DRIFTS, as well as DFT simulations, and all conclusions are obtained based on the

experimental data, we hope you can understand the Mössbauer spectroscopy cannot be carried out because of the objective conditions. Your suggestion is indeed helpful, and we will take the Mössbauer spectroscopy characterization in our further work.

Q4: What will happen when Pd-Fe₃O₄-Re is reduced in H₂ at 300 °C for 1 h and whether there is some difference between the structure of resultant and Pd-Fe₃O₄-H.

Response: Thanks for the question. We treated the Pd-FeO_x NPs in H₂ to obtain Pd-Fe₃O₄-H and SMSIR was formed. In the previous version, we have further treated the Pd-Fe₃O₄-H in air to check the reversibility of SMSIR. As shown in the TEM image of the Pd-Fe₃O₄-Re (**Figure A8**), the sample still possesses a yolk-shell structure, but the voids are smaller than those of Pd-Fe₃O₄-H. The XPS of Pd-Fe₃O₄-Re (**Figure A9**) demonstrates three Pd states of metallic Pd, Pd^{δ+} in Pd-Fe bond, and PdO. The intensity of Pd^{δ+} peak is lower than that of Pd-Fe₃O₄-H (**Figure A3**), indicating the decrease of Pd^{δ+} concentration. The CO DRIFTS of Pd-Fe₃O₄-Re in **Figure A9** shows that the intensity of CO-Pd^{δ+} became weaker than that in the CO DRIFTS of Pd-Fe₃O₄-H in Supplementary **Figure A4**, suggesting the CO DRIFTS spectral feature is an intermediate state between Pd-Fe₃O₄-H and Pd-Fe₃O₄-A. The analysis of TEM, CO DRIFTS and XPS together suggests that the SMSIR in this work is partially reversible.

We understand that the reviewer is curious about what will happen when Pd-Fe₃O₄-Re is reduced again in H₂ at 300 °C for 1 h and whether there is some difference between the structure of resultant and Pd-Fe₃O₄-H. With the fact that when the Pd-Fe₃O₄-H was treated in air, the structure is partially revisable, we anticipate that if the Pd-Fe₃O₄-Re is further retreated in hydrogen, the structure can not to be totally recover to the same structure of Pd-Fe₃O₄-H, because of the different structures of Pd-Fe₃O₄-A and Pd-Fe₃O₄-Re. **The SMSIR in this work is partially revisable. Therefore, it's reasonable to assume Pd-Fe₃O₄-Re reduced in H₂ at 300 °C would partially recover to the state of Pd-Fe₃O₄-H, but not a 100% recovery.**

Figure A8 TEM image of Pd-Fe₃O₄-Re.

Figure A9 High-resolution Pd 3d XPS spectrum of Pd-Fe₃O₄-Re.

Q5: It should be “Fe₃O₄ islands on Pd” not “Pd islands on Fe₃O₄” in Q3.

Response: Thank you for pointing out our typo. We have checked our submission to make sure we do not have this typo in both manuscript and SI.

References:

- 1 Benziger, J. B. & Larson, L. R. An infrared spectroscopy study of the adsorption of CO on Fe/MgO. *J. Catal.* **77**, 550-553, doi:10.1016/0021-9517(82)90195-6 (1982).
- 2 Felicissimo, M. P., Martyanov, O. N., Risse, T. & Freund, H.-J. Characterization of a Pd-Fe bimetallic model catalyst. *Surf. Sci.* **601**, 2105-2116, doi:10.1016/j.susc.2007.02.023 (2007).
- 3 Wei, X., Ma, Z., Lu, J., Mu, X. & Hu, B. Strong metal-support interactions between palladium nanoclusters and hematite toward enhanced acetylene dicarbonylation at low temperature. *New J. Chem.* **44**, 1221-1227, doi:10.1039/C9NJ05493F (2020).

REVIEWERS' COMMENTS:

Reviewer #2 (Remarks to the Author):

The authors have clearly done their best now, the quality of the manuscript has been obviously improved, and now I can recommend accepting for publication.